# Towards Effective Planning Strategies for Dynamic Opinion Networks

**Bharath Muppasani, Protik Nag, Vignesh Narayanan, Biplav Srivastava, and Michael N. Huhns**
AI Institute and Department of Computer Science
University of South Carolina, USA
{bharath@email., pnag@email., vignar@, biplav.s@, huhns@}sc.edu

## Abstract

In this study, we investigate the under-explored *intervention planning* aimed at disseminating accurate information within *dynamic opinion networks* by leveraging learning strategies. Intervention planning involves *identifying key nodes* (search) and *exerting control* (e.g., disseminating accurate/official information through the nodes) to mitigate the influence of misinformation. However, as the network size increases, the problem becomes computationally intractable. To address this, we first introduce a ranking algorithm to identify key nodes for disseminating accurate information, which facilitates the training of neural network (NN) classifiers that provide generalized solutions for the search and planning problems. Second, we mitigate the complexity of label generation—which becomes challenging as the network grows—by developing a reinforcement learning (RL)-based centralized dynamic planning framework. We analyze these NN-based planners for opinion networks governed by two dynamic propagation models. Each model incorporates both binary and continuous opinion and trust representations. Our experimental results demonstrate that the ranking algorithm-based classifiers provide plans that enhance infection rate control, especially with increased action budgets for small networks. Further, we observe that the reward strategies focusing on key metrics, such as the number of susceptible nodes and infection rates, outperform those prioritizing faster blocking strategies. Additionally, our findings reveal that graph convolutional network (GCN)-based planners facilitate scalable centralized plans that achieve lower infection rates (higher control) across various network configurations (e.g., Watts-Strogatz topology, varying action budgets, varying initial infected nodes, and varying degree of infected nodes).

## 1 Introduction

The spread of information across social networks profoundly impacts public opinion, collective behaviors, and societal outcomes [1]. Especially during crises such as disease outbreaks or disasters, there is often too much information coming from different sources. Sometimes, the resultant flood of information is unreliable or misleading, or spreads too quickly, which can have serious effects on society and health [8]. Online platforms such as Facebook, X, and WeChat, while essential for communication, significantly contribute to the rapid spread of misinformation [2]. This has led to public confusion and panic in events ranging from the Fukushima disaster to the COVID-19 pandemic, and even the U.S. presidential elections [11], demonstrating the need for effective information management on these platforms [8].

The first step in combating misinformation propagation is to reliably detect them. However, detection alone is insufficient to effectively mitigate its spread. It must be complemented with strategic

38th Conference on Neural Information Processing Systems (NeurIPS 2024).

*intervention planning* to contain its impact. In this context, numerous studies have focused on rumor detection [53, 47, 40, 19], while comprehensive strategies for controlling misinformation are limited [15]. Existing research works on controlling misinformation emphasizes three primary strategies: node removal [9, 45, 29], edge removal [21, 19, 44], and counter-rumor dissemination [5, 42, 12]. Node removal involves identifying and neutralizing key nodes using community detection methods, with dynamic models that adapt to changes in propagation. For example, [50, 16] present ranking algorithms that are critical for identifying influential nodes within complex networks, which can then be targeted to block, remove, or cascade information, to reduce the overall spread of misinformation. These algorithms rank nodes based on various metrics, determining their importance or influence within the network. The second approach, edge removal, focuses on disrupting misinformation pathways by strategically severing connections between nodes. For example, authors in [38] considered mitigating misinformation by identifying potential purveyors to block their posts. However, taking strong measures like censoring user posts may violate user rights.

The third strategy, counter-rumor dissemination, promotes the spread of factual information, leveraging user participation and 'positive cascades' to counteract misinformation [41]. For instance, authors in [13] developed a method that involves learning an intervention model to enhance the spread of true news, thereby reducing the influence of fake news. The effectiveness of such intervention planning methods relies on the ability of intervention models to identify key nodes and disseminate accurate/official information through these nodes to mitigate the influence of misinformation. Following this strategy, in [34], a search problem was formulated and sequential plans using *automated planning* techniques were generated for the targeted spread of information. Despite several model-based efforts to strengthen this approach (see Appendix A.1 for a review of relevant literature), the existing research often overlooks key features of opinion propagation models, such as their rich network dynamics, asynchronous communication, and the impact of factors like the degree of infected nodes, action budget, and various reward models on the effectiveness of planners.

We address this gap by studying the intervention planning problem using two learning methodologies. In both these methodologies, we investigate three distinct cases of opinion network models, ranging from discrete to continuous representations of opinion and mutual trust. Specifically, in this paper, we propose a novel ranking algorithm integrated with a supervised learning (SL) framework to identify influential nodes using a robust feature set of network nodes and evaluate their performance in all three cases. Additionally, we also develop a reinforcement learning (RL)-based solution to design centralized planners that are suitable for larger networks. Furthermore, we develop comprehensive datasets with a wide range of Watts-Strogatz (or small-world) network topologies, varying degrees of initial infected nodes, action budgets, and reward models, e.g., from those requiring local network information to those utilizing global real-time network states, and analyze both the developed methodologies. Next, we use an example to illustrate the intervention planning problem and highlight our contributions.

**Example:** Consider a research community discussing the NeurIPS submission deadline. In this context, a *topic* is just a statement such as 'NeurIPS submission deadline is on May 22'. An *opinion* of an agent on this topic is defined as the belief of the agent in the truthfulness of the statement [3]. A positive (respectively negative) opinion value represents that the agent believes the statement is true (respectively false). In our study, we consider the opinion value of an agent on a topic lies in $[-1, 1]$. Our problem setup consists of a network of connected agents (with opinion values on a given

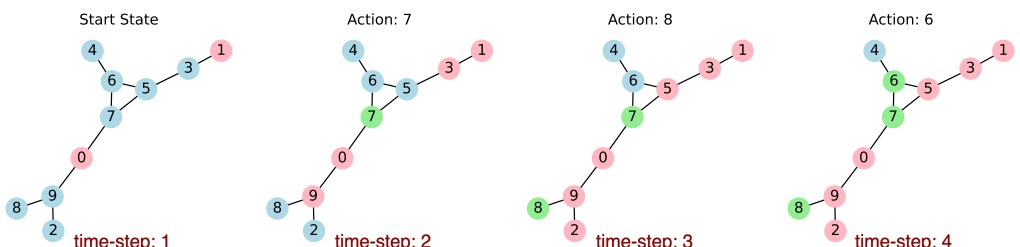

Figure 1: Example of misinformation propagation and control choices at each timestep. Blue nodes: neutral (opinion value 0), red nodes: misinformed (opinion value $-1$), green nodes: received accurate information (opinion value 1).

topic) represented by a graph. We consider that a subset of agents propagate misinformation to their neighbors. Our goal is to counteract this by disseminating accurate information to selected agents at each time step. The agents receiving accurate information update their opinion to a level where they no longer believe the misinformation and, consequently, cease to propagate it. The process ends when no agents are left to receive the misinformation. Figure 1 illustrates such a scenario where the red nodes represent the agents not believing about the NeurIPS deadline being May 22. When they interact with other agents, they share this misinformation, leading to a spread of incorrect information throughout the network. The blue nodes represent neutral agents who are unaware of the deadline, and the green nodes represent agents who believe in the NeurIPS deadline being on May 22. Initially, agents, i.e., nodes $\{0, 1\}$, propagate misinformation to their neighbors. To control the spread of this misinformation, timely control actions are taken at each timestep to disseminate accurate information to selected agents. In the example, with an action budget of $1$, agents $7$, $8$, and $6$ are sequentially chosen to receive the accurate information, represented by the green nodes. Through these control actions, the example demonstrates how timely intervention at critical points can effectively mitigate misinformation spread, ensuring agents are accurately informed.

Our paper makes several significant **contributions** to intervention planning, focusing on the integration of more realistic and complex modeling approaches, label generation techniques, and training methodologies: (a) We utilize a continuous opinion scale to model the dynamics of misinformation spread (instead of just binary scale as shown in the example), providing a more realistic representation of opinion changes over time. (b) We develop a ranking algorithm for generating labels in networks with discrete opinions, addressing a significant gap in efficient data preparation for SL algorithms in this domain. (c) We develop an RL methodology and perform comprehensive analysis. This allows for adaptive intervention strategies in response to evolving misinformation spread patterns, a critical improvement over traditional static approaches. (d) Utilizing graph convolutional networks (GCNs) with an enhanced feature set of opinion value, connectivity degree, and proximity to a misinformed node, we improve the training of models for selecting effective intervention strategies. This enhancement ensures scalability and generalizes well across various network structures, demonstrating the robust capabilities of GCNs in complex scenarios. Table 3 highlights our contributions by providing a comparative overview of the key features and innovations in our work to that of previous studies. The code and the datasets developed as part of the analysis presented in this paper can be found in [33].

## 2 Problem Formulation

In this section, we discuss our approach to modeling the opinion network environment, the dynamics of information propagation, and our strategy for containing misinformation.

### 2.1 Environment Description

An opinion network is formally represented as a directed graph $G = (V, E)$, where $V$ denotes the set of nodes (agents), and $E$ denotes the set of edges (relationships or trust) between agents [15]. The graph structure we consider for our study is undirected, indicating that relationships between agents are bi-directional. Each node within the graph represents an individual agent, and each agent holds a specific opinion on a given topic. An edge between any two nodes signifies a direct connection or relationship between those agents, facilitating the exchange of opinions.

Opinion values are quantified within the range $[-1, 1]$, representing different levels of belief on a given topic, and the weight assigned to each edge quantifies the mutual trust level between connected agents, scaled within the interval $[0, 1]$. While existing works in the literature have explored only binary opinion and trust models, in computational social science, researchers have developed models with opinion and trust values as continuous variables. Investigation of planning strategies in continuous models remains under-explored. In this paper, we explore three distinct cases of opinion and trust values. `Case-1` involves binary opinion values with binary trust, simplifying the network dynamics into discrete states. In `Case-2`, we use floating-point opinion values while maintaining binary trust, allowing for a more granular assessment of opinions while still simplifying trust dynamics. Finally, `Case-3` features both floating-point opinion values and floating-point trust, representing more realistic opinion and trust relationships within the network capturing continuous variations.

## 2.2 Propagation Model

In the analysis of opinion networks, it is essential to understand how opinions form and evolve, guided by the dynamics of trust among agents. In our analysis, the evolution and propagation of opinions within opinion networks are modeled using a linear adjustment mechanism (*discrete linear maps*), as described by the following transition function

$$x_i(t+1) = x_i(t) + \mu_{ik}(x_k(t) - x_i(t)), \quad t = 0, 1, \ldots. \tag{1}$$

Equation 1 models the dynamics of opinion evolution, where the opinion of agent $i$ at time $t + 1$ depends on its current opinion and the influence exerted by a connected agent $k$, who is actively sharing some information with agent $i$, moderated by the trust factor $\mu_{ik}$. This asynchronous propagation model adapts differently across various experimental setups as detailed below.

In `Case-1` and `Case-2`, where mutual trust values are discrete $\{0, 1\}$, the application of Equation 1 results in immediate shifts in opinion. For example, if an agent $i$ with a current opinion value of 0.5 on some topic is influenced by a connected neighboring agent $k$ with an opinion value of -1 on the same topic, agent $i$'s opinion immediately shifts to -1 in the next timestep, reflecting a discrete transition. Conversely, in `Case-3`, which involves a continuous range of opinion and trust values, changes are more gradual. Here, if agent $i$ holds an opinion of 0.5 and is influenced by a neighbor $k$ with an opinion of -1 and a moderate trust factor, the opinion of agent $i$ incrementally moves closer to -1 in subsequent timesteps. This reflects a gradual shift towards a consensus opinion, depending on the magnitude of the trust level between agents $i$ and $k$.

To further enhance our understanding of opinion dynamics in networks with continuous trust relationships, we have also used the DeGroot propagation model [6] in `Case-3`. The propagation of opinions in this model is governed by the following equation:

$$x_i(t+1) = \sum_{k=1}^{n} \mu_{ik} x_k(t), \quad t = 0, 1, \ldots. \tag{2}$$

Equation 2 describes the opinion of agent $i$ at time $t + 1$ as a weighted average of the opinions of all the neighboring agents at time $t$, where the weights $\mu_{ik}$ represent the trust agent $i$ has in agent $k$. Often, in the DeGroot model, which is a synchronous propagation model, the summation in 2 is a *convex sum*, i.e., the trust values add to one so that we have $\sum_{k=1}^{n} \mu_{ik} = 1$ for each $i = 1, \ldots, n$. This normalization allows the DeGroot model to exhibit stable asymptotic behaviour.

At each timestep, the following processes occur: Nodes with opinion values lower than -0.95 are identified as sources of misinformation and transmit the misinformation to their immediate neighbors (referred to as 'candidate nodes') according to one of the propagation model detailed in Equations 1 and 2. Concurrently, an intervention strategy is applied where a subset of these neighbors—constrained by *an action budget*—is selected to receive credible information from a trusted source. This source is characterized by an opinion value of 1 and we vary trust parameter among 1, 0.8, and 0.75. The process includes a blocking mechanism where a node that exceeds a positive opinion threshold of 0.95 is considered 'blocked', ceasing to interact with the misinformation spread or disseminate positive influence further. The simulation concludes when there are no viable 'candidate nodes' left to propagate misinformation. **Our primary objective** is to devise a learning mechanism that efficiently identifies and selects key nodes within the network to disseminate accurate information at each time step.

## 3 Methods

In this section, we will explain our methodologies, presenting an overview of the network architectures employed, including the GCN and ResNet frameworks. Additional details about the neural network architecture utilized for our experiments can be found in Appendix A.2. We detail our proposed ranking algorithm utilized in the SL process. Additionally, we elaborate on the implementation of the Deep Value Network (DVN) with experience replay for the proposed RL-based planners. Furthermore, we provide an explanation of the various reward functions analyzed in our RL setup.

## 3.1 Ranking Algorithm-based Supervised Learning

In this section, we propose a ranking algorithm based SL model to classify the key nodes at each time step to disseminate accurate information. Our SL method utilizes a GCN architecture.

**Ranking Algorithm:** We pose the ranking algorithm as a search problem where the objective is to find the optimal set of nodes that, when blocked, minimizes the overall infection rate. The network is represented as a graph $G$, where nodes can be infected, blocked, or possess opinion values within the range $[-0.95, 0.95]$. Initially, a simulation network $S$ is created by setting the opinion values of infected nodes to $-1$ and removing blocked nodes. Let $M$ denote the number of nodes in $S$ that are neither infected nor blocked. Given an action budget $K$, we select $K$ nodes from $M$ in $\binom{M}{K}$ possible ways, forming the set $C$ of all possible combinations. For each subset $c \in C$, we temporarily block the nodes in $c$ by setting their opinion values to $1$ and simulate the spread of misinformation within $S$. The resulting infection rates for each subset $c$ are stored in the set $R$. We identify the subset $c^* \in C$ that yields the minimal infection rate, denoted as $r^*$. This subset $c^*$ is our target set. We then construct a target matrix $T \in \mathbb{R}^{N \times 1}$, where $N$ is the total number of nodes in the original network $G$. All entries of $T$ are initialized to 0, and for each node $i \in c^*$, the $i$-th entry of $T$ is set to 1. This target matrix $T$ is subsequently used to train the GCN-based model. A pseudocode for this ranking algorithm is presented in Algorithm 1 in Appendix A.5.

**Overall Training Procedure:** The training of our GCN-based model leverages the labels defined in the target matrix $T \in \mathbb{R}^{N \times 1}$. This matrix is compared with the model's output matrix $O \in \mathbb{R}^{N \times 1}$, which estimates the blocking probability of each node. We evaluate training efficacy using the binary cross-entropy loss between $T$ and $O$, which quantifies prediction errors. Model weight adjustments are implemented via standard backpropagation [23] based on this loss.

Each training iteration consists of several episodes, starting with the generation of a random graph state $G$ containing initially infected nodes. The GCN then processes this state to output matrix $O$ using the graph's features and structure. Labels are generated, as detailed above using the ranking algorithm, generating the target matrix $T$. The binary cross-entropy loss between $O$ and $T$ is calculated for backpropagation. The environment updates by blocking predicted nodes, allowing the infection spread, and adjusting node attributes. The process repeats until misinformation spread is halted, with each episode refining the graph's state for subsequent iterations. The results of the planners for difference cases are summarized in the Appendix A.6.1.

However, we note that, in the case of continuous opinion and continuous trust (`case-3`), the process of label generation becomes more complex. In such scenarios, agents do not change their opinion immediately but gradually, making it difficult to predict which agents will be misinformed based on a single propagation simulation. Therefore, simulations across multiple time steps are necessary to identify the optimal nodes to block. As the ranking algorithm uses a brute force method to determine optimal nodes, this approach becomes increasingly challenging with continuous opinion models.

## 3.2 Reinforcement Learning-based Centralized Dynamic Planners

In SL, the process of generating labels can be costly and impractical as network size increases. This is evident while considering mitigating misinformation propagation in large networks, where identifying the optimal set of nodes for blocking requires a combinatorial search that is computationally infeasible. Thus, RL emerges as a viable alternative.

Deep $Q$-networks (DQNs) [32] using random exploration combined with experience replay have been demonstrated to effectively learn $Q$-values for sequential decision making with high-dimensional data. Unlike the classical DQN, where the network outputs a $Q$-value corresponding to each possible action, in our problem, which also deals with high-dimensional data, we develop a DVN, as the number of available actions at each time step need not be fixed. Consequently, the output layer consists of a single neuron that outputs the value for a given input state. The agent's experiences at each time step are stored in a replay memory buffer for the neural network parameter updates. The loss function for training is given by

$$\mathcal{L}(s_t, s_{t+1}|\theta) = \left( r_t + \hat{V}_{\theta^-}(s_{t+1}) - V_\theta(s_t) \right), \tag{3}$$

where $t$ represents the current time step, $s_t$ is the current state, $s_{t+1}$ denotes the subsequent state after action $a_t$ is taken, and $r_t$ is the reward received for taking $a_t$ in $s_t$. The parameters $\theta$ denote

the weights of the value network used to estimate the state value $V_\theta(s_t)$, while $\theta^-$ represents the parameters of a target network, typically a lagged copy of $\theta$, used to stabilize training. Here, $\hat{V}_{\theta^-}(s_{t+1})$ is the estimated value of the next state $s_{t+1}$ according to the target network. The specific reward functions used in this study are discussed later in the section. Algorithm 2, in Appendix A.5, provides a detailed implementation of our DVN with experience replay.

### 3.2.1 Reward Functions for RL setup

The reward function is designed to encourage policies that effectively mitigate the spread of misinformation. Specifically, the reward functions modeled for our study are: (1) $R_0 = -(\Delta\text{infection rate})$, where $\Delta$infection rate is defined as the change in infection rate resulting from taking action $a_t$. Specifically, $\Delta$infection rate = infection rate at $s_{t+1}$ − infection rate at $s_t$. This reward structure encourages the model to reduce the rate at which misinformation spreads by penalizing increases in the infection rate; (2) $R_1 = -(\# \text{ candidate nodes})$, targets the number of immediate neighbors of infected nodes that are susceptible to becoming infected in the next timestep, thereby promoting strategies that minimize the potential for misinformation to spread; (3) $R_2 = -(\# \text{ candidate nodes}) - (\Delta\text{infection rate})$, takes into account the previous two rewards, balancing the need to control both the number of susceptible nodes and the overall infection rate; (4) $R_3 = 1 - (\frac{\# \text{ time steps}}{\text{Total time steps}})$, rewards quicker resolutions, providing higher rewards for strategies that contain misinformation rapidly and evaluating the effectiveness only at the end of each episode; (5) $R_4 = -(\text{infection rate})$, directly penalizes the current infection rate, thus favoring actions that achieve lower overall infection rates; and finally, a combined reward that incorporates elements of both $R_3$ and $R_1$. Throughout the simulation, the agent continually receives rewards based on the number of candidate nodes, fostering strategies that limit the expansion of the infection network. As the simulation concludes, the agent receives an episodic reward calculated as (6) $R_5 = -(\# \text{ candidate nodes}) - \frac{\# \text{ time steps}}{\text{Total time steps}}$, thereby reinforcing the importance of quick and efficient resolution of misinformation spread. Note that all these reward structures, in addition to differing in how they represent the goal for the planners, also differ in the network information required to compute them.

### 3.3 Network Architectures

In our experiments, we utilized two neural network architectures. First, a GCN to model node features within a network. Each node was characterized by three key features. These features were represented in a matrix $F \in \mathbb{R}^{N \times 3}$, where $N$ denotes the total number of nodes. The feature matrix is dynamic and evolves to reflect changes in the network. It includes the opinion value, the connectivity degree, which identifies nodes potentially susceptible to misinformation while excluding those already blocked or misinformed, and the proximity to a misinformed node, which is calculated as the shortest path to the nearest infected node, assigning a distance of infinity to unreachable nodes.

We have also considered using Residual Network (ResNet) Architecture. The ResNet model implemented in our study is a variant of the conventional ResNet architecture [14]. The core component of our ResNet model is the `ResidualBlock`, which allows for the training of deeper networks by addressing the vanishing gradient problem through skip connections. Each `ResidualBlock` consists of two sequences of convolutional layers (`Conv2d`), batch normalization (`BatchNorm2d`), and sigmoid activations. Complete details about the model architectures used in our study are provided in Appendix A.2.

## 4 Experiments

In this section, we present the details about training data generation and configurations chosen for our SL and RL methodologies. We also explain the test data used for evaluating the trained models.

### 4.1 Training Setup

In the SL setup, we experimented with three distinct graph structures: Watts-Strogatz (with nearest neighbors $k = 3$ and a rewiring probability $p = 0.4$), Erdos-Renyi (with branching factor of 4), and Tree graphs (with branching factors randomly selected from the range $[1, 4]$). Each graph type facilitated training models to evaluate the influence of various structural dynamics on performance.

We used the GCN model for the SL method. Due to the consistent performance of the trained models on the different graph topologies, we chose the small-world topology to present all the subsequent analyses and summarize the results in Appendix A.6.1. On the other hand, we trained centralized RL planners using both ResNet and GCN network architectures. Each trained configuration is represented as $model\text{-}n\text{-}x\text{-}y$, where $model \in \{\text{ResNet, GCN}\}$, $n \in \{10,25,50\}$ represents the network length (in terms of the number of nodes), $x \in \{1,2,3\}$ represents the number of initial infected nodes, and $y \in \{1,2,3\}$ represents the action budget.

## 4.2 Test Data Generation

The datasets used in related works, such as [17], typically consist of network structures, and no real-time opinion propagation data could be found. Therefore, to evaluate our intervention strategies, we generated two sets of synthetic datasets using the Watts-Strogatz model with the training dataset's configurations. This approach allows us to simulate complex networks and control the structure, connectivity, and initial infected nodes to assess our models effectively.

**Dataset v1** examines the effects of network size and the initial count of infected nodes on misinformation spread. We generated data with network sizes of 10, 25, and 50 nodes with 1, 2, and 3 initially infected nodes, respectively, creating 9 unique datasets. Each configuration has 1000 random network states with the opinion values of non-infected nodes uniformly distributed between $-0.5$ and $0.6$.

**Dataset v2** examines how the initial connections of infected nodes affect the spread of misinformation. Like Dataset v1, it includes networks of 10, 25, and 50 nodes. However, the initial number of connections (degrees of connectivity) for the infected nodes varies from 1 to 4. Here by degree of connectivity, we mean the number of `candidate nodes` present at the start of the simulation. This variation results in a total of 12 datasets for each configuration, with each dataset containing 1000 states. In these configurations, the number of initially infected nodes is randomly chosen from 1 to 3.

# 5 Results and Discussion

In this section, we evaluate the models, using the `infection rate` metric, trained using our ranking-based SL and RL algorithms with various reward functions. We discuss the efficiency of these models using `Dataset v2`, particularly on a network of 50 nodes with a connectivity degree of 4, as it represents the most complex test dataset we generated. Similar evaluation results for other datasets can be found in the Appendix A.6. Additionally, we have also evaluated our planning algorithms using directed and undirected real-world network models reported in the literature. These evaluations are presented in Table 2. The details of the hardware used for our experiments are provided in the Appendix A.4. Our empirical investigation yielded insightful results regarding the performance of our trained models under various training conditions. With comprehensive experimental evaluations, we were able to address the following research questions.

**Objective and Research Questions: O**: Identify the optimal combination of initially infected nodes and action budget parameters for training models to effectively control the spread of misinformation. **RQ1**: For reward functions that focus on the blocking time, does adding any other factor lead to better results? If yes, which factor? **RQ2**: Do reward functions that look at global graph information perform better than those considering local, neighboring information? **RQ3**: Does GCN offer better scalability and performance when compared with ResNet.

*O: What is the best combination of initially infected nodes and action budget parameters for training the models to control the misinformation spread?*

To examine this, we focused our analysis on the Mean Squared Error (MSE) loss plots obtained during the training phase. Figure 7 in Appendix A.6 illustrates the comparison of training loss across various network parameter settings for all considered reward types in `Case-1`, employing a ResNet model trained on a network of 50 nodes. The trend in loss convergence across episodes was found to be consistent for both the ResNet and GCN models across all cases examined. The analysis revealed that reward functions exhibiting lower and more stable loss values correlate with improved model learning performance. Our findings highlight that increasing the number of initially infected nodes typically elevates the stabilization point of MSE loss, indicating a more challenging learning environment. Additionally, a higher action budget contributes to increased MSE variability, reflecting

Table 1: Inference results on Dataset v2, with a degree of connectivity 4, featuring a network of 50 nodes. This table presents the average infection rates for different models, with ResNet trained on a network of 50 nodes and GCN trained on networks of 10 nodes, under various methods (`M.`) tested with varying action budgets (`A.`) across different cases considered.

| A. | M. | Case-1 | | Case-2 | | Case-3 | |
|---|---|---|---|---|---|---|---|
| | | **ResNet(50)** | **GCN(10)** | **ResNet(50)** | **GCN(10)** | **ResNet(50)** | **GCN(10)** |
| | $RL+R_0$ | 0.2334 | 0.2481 | 0.2496 | 0.2454 | 0.0449 | 0.0461 |
| | $RL+R_1$ | 0.1917 | 0.1608 | 0.1965 | 0.1607 | 0.0449 | **0.0435** |
| | $RL+R_2$ | 0.2427 | 0.1608 | 0.2148 | 0.1608 | 0.0451 | 0.0438 |
| 1 | $RL+R_3$ | 0.2331 | 0.2958 | 0.2281 | 0.3381 | 0.0444 | 0.046 |
| | $RL+R_4$ | 0.199 | **0.1593** | 0.2513 | **0.1596** | 0.045 | 0.0442 |
| | $RL+R_5$ | - | 0.1607 | - | 0.1605 | - | 0.0439 |
| | SL+GCN(25) | 0.304 | | 0.2889 | | 0.3715 | |
| | $RL+R_0$ | 0.0974 | 0.1012 | 0.0992 | 0.1007 | **0.0398** | 0.04 |
| | $RL+R_1$ | 0.0886 | **0.0842** | 0.0901 | 0.0843 | **0.0398** | **0.0398** |
| 2 | $RL+R_2$ | 0.097 | **0.0842** | 0.0957 | 0.0843 | 0.0399 | **0.0398** |
| | $RL+R_3$ | 0.0959 | 0.0969 | 0.0962 | 0.1032 | **0.0398** | 0.04 |
| | $RL+R_4$ | 0.0898 | **0.0842** | 0.1005 | **0.0842** | 0.0399 | **0.0398** |
| | $RL+R_5$ | - | **0.0842** | - | **0.0842** | - | **0.0398** |
| | SL+GCN(25) | 0.1464 | | 0.1032 | | 0.3491 | |
| | $RL+R_0$ | 0.0599 | 0.0599 | 0.0599 | 0.06 | **0.0397** | 0.0399 |
| | $RL+R_1$ | 0.0599 | **0.0597** | 0.0598 | **0.0597** | 0.0398 | **0.0397** |
| 3 | $RL+R_2$ | 0.0598 | **0.0597** | 0.0599 | **0.0597** | 0.0398 | **0.0397** |
| | $RL+R_3$ | 0.06 | 0.0598 | 0.0601 | 0.0602 | **0.0397** | 0.0399 |
| | $RL+R_4$ | 0.0598 | **0.0597** | 0.06 | **0.0597** | 0.0398 | 0.0398 |
| | $RL+R_5$ | - | **0.0597** | - | **0.0597** | - | 0.0398 |
| | SL+GCN(25) | 0.0488 | | 0.0559 | | 0.2526 | |

the added complexity and generally poorer performance during training. Based on this analysis, we find ResNet-$n$-1-1 and GCN-$n$-1-1, $n \in \{10,25,50\}$, to be the best training configurations.

Table 1 presents the average infection rate values across different cases considered for `Dataset v2` with a degree of connectivity 4, featuring a network of 50 nodes, detailing the average infection rates. It compares the performance of the ResNet model, trained on a network of 50 nodes, with the GCN model, trained on a network of 10 nodes, using the RL training algorithm across the different reward types, and the GCN model trained using SL on a network of 25 nodes. Results on the additional datasets are provided in Appendix A.6.

*RQ1: For reward functions that focus on blocking time, does adding any other factor lead to better result? If yes, which factor?* Answer: Yes. `#candidate nodes`.

Reward function $R_3$, which is formulated to minimize the number of time steps required to halt the spread of misinformation, might inadvertently not be the most effective strategy for minimizing the overall infection rate within the network. As the reward is solely based on the speed of response, it does not directly account for the magnitude of the misinformation spread, that is, the number of nodes affected. Therefore, the agent may prioritize actions that conclude the propagation swiftly but do not necessarily result in the most substantial reduction in the spread of misinformation. However, our results indicate that under specific training configurations with an action budget or initial infected nodes greater than 1, the reward function $R_3$ outperforms others in maintaining lower infection rates, as shown in Figure 8 in Appendix A.6. This finding is significant since $R_3$, a sparser reward type, requires less computational effort and is independent of network observability. As the action budget increases the propagation tends to conclude in fewer timesteps thereby resulting in the RL agent receiving a higher reward in the case of $R_3$. Figure 2 shows that the RL agent trained with the $R_3$ reward function chooses actions that conclude propagation in the least time. Conversely, Figure 1 illustrates the sequence of actions chosen by an RL agent trained with the $R_1$ reward function on the same network. Although $R_1$ requires more time steps than $R_3$, it results in a lower infection rate. This can also be observed from Table 1, where the infection rate is higher for the $R_3$ reward function than

Table 2: Average infection rate values from experiments conducted on 100 random instantiations for each real-world network, each starting with 1 random initially infected node, obtained using GCN model trained on synthetic networks of 10 nodes, under various reward structures (`M.`) tested with varying action budgets (`A.`). The network properties of # nodes (V), # edges (E), and Average Degree for each network are shown in the table.

| A. | M. | Zachary's Karate Club [Undirected] [51] | Facebook [Undirected] [25] | Email [Directed] [24] | Cora [Undirected] [31] |
|---|---|---|---|---|---|
| Network Properties | | V: 34, E: 78 Avg. Deg.: 4.59 | V: 250, E: 1352 Avg. Deg.: 10.8 | V: 300, E: 2358 Avg. Deg.: 7.9 | V: 2000, E: 2911 Avg. Deg.: 2.9 |
| | RL+$R_0$ | 0.8579 | 0.4569 | 0.395 | 0.0603 |
| | RL+$R_1$ | **0.5279** | **0.4547** | **0.315** | **0.0095** |
| | RL+$R_2$ | **0.5279** | **0.4547** | 0.3169 | **0.0095** |
| 1 | RL+$R_3$ | 0.8468 | 0.511 | 0.3759 | 0.0609 |
| | RL+$R_4$ | 0.5326 | 0.4589 | 0.3183 | 0.0096 |
| | RL+$R_5$ | 0.5276 | 0.4547 | 0.316 | 0.0096 |
| | RL+$R_0$ | 0.2762 | 0.3621 | 0.2512 | 0.0166 |
| | RL+$R_1$ | 0.1641 | **0.3022** | 0.0988 | **0.002** |
| | RL+$R_2$ | 0.1641 | 0.3024 | 0.1062 | **0.002** |
| 3 | RL+$R_3$ | 0.2562 | 0.3697 | 0.1838 | 0.0142 |
| | RL+$R_4$ | **0.1582** | 0.3047 | **0.0966** | **0.002** |
| | RL+$R_5$ | 0.1641 | 0.3022 | 0.0988 | **0.002** |
| | RL+$R_0$ | 0.0926 | 0.233 | 0.103 | 0.0126 |
| | RL+$R_1$ | 0.0697 | 0.1649 | **0.0347** | **0.0017** |
| | RL+$R_2$ | **0.0662** | 0.1647 | **0.0347** | **0.0017** |
| 5 | RL+$R_3$ | 0.085 | 0.2039 | 0.0587 | 0.0052 |
| | RL+$R_4$ | **0.0662** | **0.1633** | 0.0351 | **0.0017** |
| | RL+$R_5$ | **0.0662** | 0.164 | **0.0347** | **0.0017** |

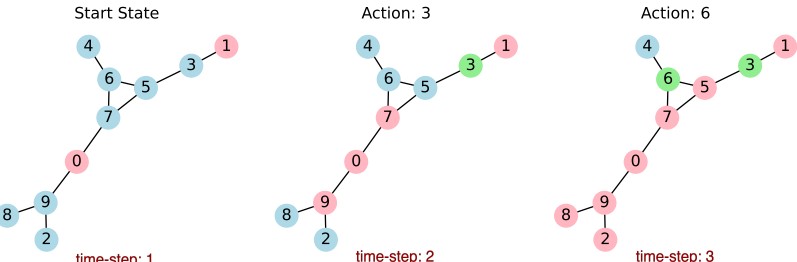

Figure 2: Sequence of actions chosen by the RL agent trained using reward function $R_3$.

any other reward function. In order to effectively implement this we have considered combining this episodic reward along with $R_1$, resulting in reward type $R_5$. This has shown a significant performance improvement when compared to the original version.

*RQ2: Do reward functions that look at global graph information perform better than those considering local, neighboring information ?* Answer: Yes

Analysis of the inference outcomes using Dataset v2, as presented in Table 1, shows that single factor reward functions, specifically $R_1 = -(\text{\# candidate nodes})$ and $R_4 = -(\text{infection rate})$, consistently resulted in lower infection rates across various settings compared to their more complex counterparts with $R_4$ providing relatively better results compared to $R_1$. This trend was observed in both ResNet and GCN models. From a practical standpoint, $R_1$ can be particularly advantageous because it does not require complete observability of the network, but just the immediate neighbors of infected nodes. Conversely, to compute $R_4$, which reflects the infection rate of the network, complete understanding of the state of each node within the network is required. This requirement for total network observability could limit the practicality of $R_4$ in situations where such detailed information is unavailable or difficult to gather.

Table 3: This table outlines the unique attributes of our approach, including the use of a deep value network, network dynamicity across multiple cases, asynchronous communication, and the exploration of five different reward models.

| Implications | Features of Our work | Our Work | Previous Work | Citations |
|---|---|---|---|---|
| Action-Space In-variant | Deep Value Net-work | ✓ | ✗ | DQN [17] |
| Expressive Models | Network Dynamic-ity | Case 1, Case 2, Case 3 | Only Case 1 | [28-40] |
| Realistic Communi-cation Dynamics | Asynchronous Com-munication | ✓ | ✗ | [37] |
| Wider Applications | Reward Models | 5 variants studies | Typically 1 | [10, 13, 17, 36] |

*RQ3: Does GCN offer better scalability and performance when compared with ResNet?* Answer: Yes

GCNs are hypothesized to outperform traditional convolution-based architectures like ResNet in tasks involving graph data due to their ability to naturally process the structural information of networks and their enhanced ability to represent complex feature sets [4]. This study compares the scalability and performance of a GCN, which excels in node classification within graphs, to a ResNet model that, despite its success in image recognition, may not scale as effectively to larger graph structures beyond the size it was initially trained on. Referring to Table 1, the GCN model, trained on only 10 node networks, consistently exhibits lower average infection rates across all the cases and under varying action budgets, when compared with the ResNet model trained on 50 node networks. The ability of GCN to maintain lower infection rates even as network complexity increases underscores its robustness and scalability in more complex network scenarios. This performance contrast highlights the suitability of GCN architectures for graph-based tasks.

# 6   Conclusions

This paper investigates scalable and innovative intervention strategies for containing the spread of misinformation within dynamic opinion networks. Our significant contributions include analysis using continuous opinion models, a design of ranking algorithm for identifying key nodes to facilitate SL-based classifiers, and the utilization of GCNs to optimize intervention strategies. Additionally, we design and study various reward functions for reinforcement learning, enhancing our approach to misinformation mitigation.

Despite significant progress, our work has limitations. In the field of computational social science, often more complex agent models are being investigated. While we have made significant efforts to extend the understanding of planning strategies, especially in continuous opinion networks, exploring complex agent traits such as stubbornness and the representation of directed trust, and implementing topic-dependency in a multi-topic network along with distributed planners instead of centralized planners as in our work is a compelling future direction.

Broader Societal Impact: This work provides methods that can be used to exert control on information spread. When used responsibly by authorized information providers, which the authors support, it will help reduce prevalent *infodemics* in social media. But it may also be misused by an adversary to wean control from an authorized party (e.g., information owner) and counter efforts to tackle misinformation. Overall, the authors believe more research efforts are needed to understand opinion networks and information dissemination strategies in dynamic and uncertain environments in pursuit of long-term societal benefits.

## Acknowledgments and Disclosure of Funding

This material is based upon work supported in parts by the Air Force Office of Scientific Research under award number FA9550-24-1-0228 and NSF award number 2337998. Any opinions, findings, conclusions, or recommendations expressed here are those of the authors and do not necessarily reflect the views of the sponsors. The authors thank the anonymous reviewers for their insightful and constructive reviews.

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

# A Appendix

## A.1 Related Works

This section reviews existing studies on controlling the flow of misinformation in networks. While most previous research has focused on detecting fake news through various features such as linguistic, demographic, or community-based indicators, there has been comparatively less work on mitigating misinformation. Mitigation strategies are typically categorized into three main approaches: removing critical nodes, severing essential connections, and countering rumors with factual information. In the following sections, we will first discuss the literature on misinformation detection, followed by a review of studies aimed at mitigating the spread of misinformation. Finally, we will provide an overview of approaches utilizing Graph Neural Networks (GNN) and Reinforcement Learning (RL) to mitigate misinformation spread.

**Misinformation Detection:** Initial efforts in misinformation detection aimed at curbing rumor spread through strategic node blocking. Wu et al. [47] developed a community detection algorithm to segment network nodes, evaluate their influence, and block key nodes. Zheng and Pan [53] addressed the least cost rumor community blocking (LCRCBO) using a community-centric influence model and a greedy algorithm to select optimal nodes for containment. However, both methods have raised concerns regarding cost-effectiveness and operational efficiency. Expanding on node-centric approaches, Ding et al. [7] developed a dynamic rumor propagation model with algorithms to identify and remove critical nodes and connections, introducing an 'outbreak threshold' to evaluate interventions. In contrast, Khalil et al. [19] and Yan et al. [49] advanced edge removal strategies under a linear threshold (LT) model, creating heuristic algorithms to manage misinformation spread effectively [18]. Tong and Du [43] used a hybrid sampling method, which could assign high weights to users susceptible to misinformation, to pinpoint users most vulnerable to fake news, while Zareie and Sakellariou [52] introduced a passive edge-blocking technique that leverages entropy to balance network diffusion efficiency. Nguyen et al. [35] applied network analysis to explore sentiment propagation in social networks, finding that sentiments often cluster within comment threads, suggesting that online forums may serve as echo chambers that reinforce uniform opinions among participants.

**Misinformation Propagation Minimization:** This research category promotes disseminating truthful information as a counter-rumor measure. Lin et al. [26] suggested a crowdsourcing framework to enable users to select from various collaborative or independent rumor control methods. Tong et al. [41] analyzed the effectiveness of the peer-to-peer independent cascade (PIC) model in private social networks, where independent rumor agents create 'positive cascades', and demonstrated that such strategies are robust under Nash equilibria. Yan et al. [48] identified the rumor minimization challenge as monotonically decreasing and devised a two-stage process for selecting effective blocking candidates. Manouchehri et al. [30] addressed maximizing influence blocking (IBM) with considerations for timing and urgency, proposing an efficient, theoretically sound sampling method. Lastly, Srinivasan and LD [39] proposed a competitive cascade model that focuses on leveraging user opinions and the critical nature of rumors to identify and activate influential nodes promoting positive information.

These studies highlight the complexity of misinformation propagation minimization and underscore the need for deeper analysis. Our study builds on these works by exploring how misinformation spreads under different environmental conditions and agent dynamics. We aim to propose more effective mitigation strategies, contributing to a nuanced understanding of misinformation dynamics and enhancing the resilience of information networks.

**GNN Based Approaches:** Graph Convolutional Networks (GCNs) are increasingly being utilized to detect the rumors and propagation patterns within social networks. For instance, Wei et al. [46] developed a GCN-based model to analyze user stances and conversation content for better rumor detection. Ma et al. [28] created a tree kernel to compare similarities between subtrees in retweeting trees. Kumar and Carley [22] employed a multitask learning framework to extract representations from retweeting trees for rumor and stance detection. Similarly, Khoo et al. [20] explored various influences within a retweeting tree and utilized a transformer model to enhance rumor detection by learning the interactions among these influences. Moreover, Liu et al. [27] proposed a structure-aware

retweeting GNN that identifies rumor patterns based on retweeting behaviors. This method leverages both node and structural-level data, suggesting that propagation paths offer distinct insights into the credibility of the disseminated information. Contrastingly, while the detection of rumors has been extensively studied, the use of GCNs for rumor minimization remains under-explored. Authors in [15] introduce an innovative approach for blocking rumors on social networks by integrating user opinions with confidence levels into a new model (CBOA) and employing a directed GCN (DGCN) to identify and block critical nodes capable of mitigating rumor spread.

However, there are opportunities to enhance their study. Our research investigates the scalability of GCNs and their performance across three different environments with varying agent dynamics. Additionally, we propose a novel ranking algorithm for training GNNs. Furthermore, we identify scenarios where supervised learning faces challenges and address these limitations using reinforcement learning techniques.

**RL** Goindani and Neville [13] develop a social reinforcement learning approach to mitigate the spread of fake news through social networks. Their method involves learning an intervention model to enhance the spread of true news, thereby reducing the influence of fake news. The authors in [10] model the news diffusion process using a Multivariate Hawkes Process (MHP) and employ policy optimization to learn intervention strategies. Ohi et al. [36] investigate strategies to mitigate the spread of pandemics using a reinforcement learning approach. The model is based on the SEIR (Susceptible-Exposed-Infectious-Recovered) compartmental model. It allows for dynamic interaction where individuals move randomly, influencing the spread of the disease. The agent is trained to determine optimal movement restrictions (from no restrictions to full lockdowns) to minimize disease spread while considering economic factors.

Current research in this field largely focuses on identifying misinformation or removing nodes and edges to limit rumor propagation. While some studies, such as those by He et al. [15], investigate misinformation suppression methods, they often do not address complex environmental scenarios, which our study aims to explore. Our research specifically targets the minimization of misinformation spread after the detection of misinformed agents is done. We aim to hinder their attempts to disseminate false information by strategically blocking selective nodes with positive information. Our work begins with the implementation of a GCN-based supervised learning model to detect misinformation. To overcome the limitations of supervised learning, we further incorporate a reinforcement learning paradigm. This progression enables us to develop and optimize more effective strategies within complex network environments, ultimately enhancing the robustness of misinformation control measures.

## A.2 Model Details

### A.2.1 Graph Neural Networks (GNNs)

Graph Neural Networks (GCNs) are advanced deep learning models tailored for handling data with a graph structure. Such models are particularly adept at processing information within complex networks by learning to synthesize node representations. These representations are derived through the aggregation and transformation of information from neighboring nodes. Specifically, the representation of a node $v_i$ in a graph $G$ is iteratively updated by integrating information from its immediate neighbors, denoted as $N(v_i)$. This process employs a propagation function $f$, parameterized by neural network weights $W$, and an activation function $\sigma$. The updated representation of a node in a multi-layered GCN, as proposed in the literature, can be mathematically expressed as:

$$H^{(l+1)} = \sigma \left( \tilde{D}^{-\frac{1}{2}} \tilde{A} \tilde{D}^{-\frac{1}{2}} H^{(l)} W^{(l)} \right) \tag{4}$$

Here, $\tilde{A} = A + I_N$ represents the augmented adjacency matrix of the graph $G$, incorporating self-connections by adding the identity matrix $I_N$. The diagonal matrix $\tilde{D}_{ii} = \sum_j \tilde{A}_{ij}$ facilitates the normalization of $\tilde{A}$. The term $W^{(l)}$ denotes the weight matrix at layer $l$, and $\sigma(\cdot)$ is the activation function. The matrix $H^{(l)} \in \mathbb{R}^{N \times D}$ encapsulates the node features at layer $l$.

We engage with a graph comprising $N$ nodes in the application discussed. Our GCN outputs a vector $O \in \mathbb{R}^{N \times 1}$, representing the likelihood of each node $v_i \in V$ being pivotal for propagation through

the network. This mechanism enables the model to figure out significance of each node in mitigating information spread within the graph structure.

**GCN Architecture Overview**  Our model is structured as follows: It comprises three graph convolutional layers followed by a linear layer. The architecture is designed to progressively transform the input node features into a space where the final classification (e.g., determining the likelihood of a node being pivotal in propagation minimization).

- **Initial Graph Convolution:** The model begins with a graph convolutional layer (`GCNConv`), taking `input_size` features and transforming them into a hidden representation of size `hidden_size`. This layer is followed by a ReLU activation function.

- **Hidden Graph Convolutional Layers:** After the initial layer, the architecture includes four `GCNConv` layers, all utilizing `hidden_size` units. These layers are designed to iteratively process and refine the features extracted from the graph's structure. Each of these layers is followed by a ReLU activation to introduce non-linearity.

  - The model employs repeated application of the `GCNConv` layer with `hidden_size` units, demonstrating the capacity to deepen the network's understanding of the graph's topology through successive transformations.

- **Output Graph Convolution:** A final `GCNConv` layer reduces the hidden representation to the desired `num_classes`, preparing the features for the prediction task.

- **Linear Layer and Sigmoid Activation:** The architecture concludes with a linear transformation (`nn.Linear`) directly mapping the output of the last `GCNConv` layer to `num_classes`. A sigmoid activation function is applied to this output, producing probabilities for each class in a binary classification scenario.

In this specific implementation, the `input_size` is set to 3, indicative of the initial feature dimensionality per node. The `hidden_size` is configured at 128, providing a substantial feature space for intermediate representations. Lastly, the `num_classes` is established at 1, signifying only one numerical output for each nodes.

**Packages Used**  The development of our supervised learning models, particularly those utilizing graph convolutional networks, leveraged several Python packages instrumental in defining, training, and evaluating our models. Below is a list of these packages and a brief description of their roles in our implementation:

- `torch`: Serves as the foundational framework for constructing and training various neural network models, including those for graph-based data.

- `torch_geometric`: An extension of PyTorch tailored for graph neural networks. It provides efficient data structures for graphs and a collection of methods for graph convolutional operations, making it essential for implementing graph convolutional neural networks (GCNs).

- `networkx`: Utilized for generating and manipulating complex networks. In our project, `networkx` is primarily used for creating synthetic graph data and for preprocessing tasks that require graph analysis and manipulations before feeding the data into the neural network models.

### A.2.2  Residual Network (ResNet)

The Residual Network (ResNet) model implemented in our study is a variant of the conventional ResNet architecture. The core component of our ResNet model is the **ResidualBlock**, which allows for the training of deeper networks by addressing the vanishing gradient problem through skip connections. Each **ResidualBlock** consists of two sequences of convolutional layers (`Conv2d`), batch normalization (`BatchNorm2d`), and sigmoid activations. A distinctive feature is the adaptation of the skip connection to include a convolutional layer and batch normalization if there is a discrepancy in the input and output channels or the stride is not equal to one.

**ResNet Architecture Overview**

- **Initial Convolution:** Begins with a convolutional layer applying 64 filters of size 3x3, followed by batch normalization and sigmoid activation.

- **Residual Blocks:** Three main layers (`layer1`, `layer2`, `layer3`) constructed with the `_make_layer` method. Each layer contains a sequence of **ResidualBlocks**, with channel sizes of 32, 64, and 128, respectively. The number of blocks per layer is determined by the `num_blocks` parameter.

  - Each **ResidualBlock** implements two sequences of convolutional operations, batch normalization, and sigmoid activation, with an optional convolution in the shortcut connection for channel or stride adjustments.

- **Adaptive Input Reshaping:** Inputs are dynamically reshaped to a square form based on the square root of the second dimension, ensuring compatibility with different input sizes.

- **Pooling and Output Layer:** Concludes with an average pooling layer to reduce spatial dimensions, followed by a fully connected layer mapping 128 features to a single output, thus producing the final prediction.

**Training Details**   Our model is trained using the PyTorch library, leveraging its comprehensive suite of neural network tools and functions. The optimizer of choice is the Adam optimizer (`torch.optim.Adam`), selected for its adaptive learning rate properties, which helps in converging faster. The learning rate was set to 0.0005, balancing the trade-off between training speed and the risk of overshooting minimal loss values. The training process involved the iterative adjustment of weights through backpropagation, minimizing the loss calculated at the output layer. This procedure was executed repeatedly over batches of training data, with the model parameters updated in each iteration to reduce the prediction error.

**Packages Used**   The implementation of our ResNet model and the training process was facilitated by the following Python packages:

- `torch`: Provides the core framework for defining and training neural networks.

- `torch.nn`: A submodule of PyTorch that contains classes and methods specifically designed for building neural networks.

- `torch.nn.functional`: Offers functional interfaces for operations used in building neural networks, like activations and pooling functions.

- `torch.optim`: Contains optimizers such as Adam, which are used for updating model parameters during training.

## A.3   Metrics

**Training Loss**

- **SL**: In the Supervised Learning (SL) framework, for our study, we employed the Binary Cross Entropy (BCE) loss to train the Graph Convolutional Network (GCN) model. Here, we delineate the iterative process used during the training phase under this setting. The model is fed with a new graph state at the beginning of each iteration. The GCN produces an output vector $O \in [0, 1]^{N \times 1}$, where $N$ is the number of nodes in the graph. Each component of O, denoted as $O_i$, represents the probability that node $i$ should be blocked to minimize the propagation rate in the network. A greedy algorithm is employed to ascertain the optimal node to block. For each node $i$, temporarily set the node as blocked. We compute the propagation rate of the network with node $i$ blocked. Then, we revert the blockage of node $i$ and proceed to evaluate the next node. We select the node that, when blocked, results in the lowest propagation rate across the network. Upon determining the node $j$, which yields the minimum propagation rate when blocked, a target vector $T \in {0, 1}^{N \times 1}$ is constructed such that $T_j = 1$ (indicating the target node to block), and $T_i = 0$ for all $i \neq j$.

  The BCE loss between the output vector $O$ and the target vector $T$ is computed as follows:

$$\text{BCE Loss} = -\frac{1}{N} \sum_{i=1}^{N} [T_i \log(O_i) + (1 - T_i) \log(1 - O_i)]$$

The Binary Cross Entropy (BCE) loss is particularly useful in this context because it measures the performance of the model in terms of how effectively it can predict the binary outcome (block/no block) for each node in the graph. Unlike our reinforcement learning setup, which utilizes batch updates across multiple states or episodes, the supervised learning approach updates the model weights based on the loss calculated from a single graph state per iteration.

- **RL**: The training loss for our model is computed using the Mean Squared Error (MSE) metric. In each episode, the loss is calculated across a batch of samples drawn from the replay buffer. Specifically, it measures the squared difference between the predicted value function of the current state from the policy network and the Bellman target — the observed reward plus the value of the subsequent state as estimated by the target network. Executing this calculation over batches of experiences allows the policy network to learn from a diverse set of state transitions, thereby refining its predictions to better approximate the true expected rewards through temporal difference learning.

**Evaluation Metric - Infection Rate**    The infection rate measures the proportion of the network that is infected over time and is a crucial metric for evaluating the spread of misinformation within a simulated environment. It is calculated as the ratio of infected nodes to the total number of nodes within the network at a given timestep:

$$\text{Infection Rate} = \frac{\text{Number of Infected Nodes}}{\text{Total Number of Nodes}}$$

This metric serves not only as a means to understand the dynamics of misinformation spread during simulations but also as a vital testing metric for evaluating model performance on test datasets. Models that effectively contain or reduce the infection rate are considered to have performed well, as they demonstrate the ability to mitigate the spread of misinformation across the network.

### A.4    Hardware details

We have used two servers to run our experiments. One with 48-core nodes each hosting 2 V100 32G GPUs and 128GB of RAM. Another with 256-cores, eight A100 40GB GPUs, and 1TB of RAM. The processor speed is 2.8 GHz.

### A.5    Training Details

#### A.5.1    SL

**Training Methodology:**    Our SL setup is coupled with a ranking algorithm which is shown in Algorithm 1. We GCN with an input size of 3 (opinion value, degree of node, proximity to source node), a hidden size of 128, and an output size of 1. The model was trained using the `Adam` optimizer with a learning rate of 0.001 and a binary cross-entropy loss function. The training process involved 1000 epochs, where in each epoch, a graph with 25 nodes was generated. During each epoch, the model iteratively minimized the infection rate by selecting nodes to block based on GCN output until no uninfected nodes remained. The loss was calculated and backpropagated, and the weights were updated accordingly. The total loss for each epoch was averaged over iterations.

#### A.5.2    RL

**Training Algorithm:**    Shown in Algorithm 2.
In the DQN framework with experience replay, two neural networks are utilized: the target network $\hat{Q}_{\theta^-}$ and the policy network $Q_\theta$. The target network is periodically updated by copying weights from the policy network, while the policy network is optimized continuously through the replay memory $D$, which stores agent's past experiences. During training, the agent samples random mini-batches of transitions from $D$ to minimize the loss function via gradient descent, enabling the policy network

---
**Algorithm 1** Ranking Algorithm
---
1: **Input:** Graph $G$, infected nodes $I$, blocked nodes $B$, action budget $K$
2: **Output:** Target matrix $T \in \mathbb{R}^{N \times 1}$
3: $S \leftarrow$ initialize_simulation_network$(G, I, B)$
4: $M \leftarrow$ get_uninfected_and_unblocked_nodes$(S)$
5: $C \leftarrow$ combinations$(M, K)$
6: $min\_rate \leftarrow 1$
7: $target\_set \leftarrow$ None
8:
9: **for** $c \in C$ **do**
10: $\quad$ $temp\_S \leftarrow S$
11: $\quad$ block_nodes$(temp\_S, c)$ $\qquad\qquad$ ▷ Temporarily block nodes in $c$ by setting their opinion values to 1
12: $\quad$ $rate \leftarrow$ simulate_propagation$(temp\_S)$ $\qquad$ ▷ Infection Rate $= \frac{\text{Number of Infected Nodes}}{\text{Total Number of Nodes}}$
13: $\quad$ **if** $rate < min\_rate$ **then**
14: $\quad\quad$ $min\_rate \leftarrow rate$
15: $\quad\quad$ $target\_set \leftarrow c$
16: $\quad$ **end if**
17: **end for**
18:
19: $T \leftarrow [0]^{N \times 1}$
20: **for** $i \in target\_set$ **do**
21: $\quad$ $T[i] \leftarrow 1$
22: **end for**
23:
24: **return** $T$
---

to estimate optimal actions. This strategy helps the agent break the temporal correlation inherent in sequential data, enhancing the stability and convergence of the learning process. We use the Mean Squared Error (MSE) metric to calculate the loss value between the policy network and the target network.

**Training Methodology:** The Neural Network model is trained using a variant of Q-learning, with a replay buffer approach to stabilize the learning process, aimed at learning the value function. Training commences with a fully exploratory policy (epsilon = 1) and transitions to an exploitation-focused strategy as epsilon decays over time. The learning rate is set to $5 \times 10^{-4}$, and mean squared error (MSE) loss is utilized to measure the prediction quality.

At each episode, 200 random initial states are generated, with a selected parameter of the number of initially infected nodes, and actions are determined through an epsilon-greedy method, balancing between exploration and exploitation. The agent performs actions by selecting nodes in the network to effectively contain misinformation spread. For each action, the network's new state and corresponding reward are observed, which are then stored in the replay buffer.

Batch updates are carried out by sampling from this buffer, ensuring that learning occurs across a diverse set of state-action-reward-next-state tuples. We have used a batch-size of 100 across the experiments. The policy network parameters are optimized using the Adam optimizer, and the target network's parameters are periodically updated to reflect the policy network, reducing the likelihood of divergence.

The training process continues for 300 number of episodes, with the epsilon parameter decaying after each timestep within an episode, encouraging the model to rely more on learned values rather than random actions as training progresses. The duration of each episode and overall training, along with average rewards and loss, are logged for post-training analysis. The model parameters yielding the best performance on the validation set are preserved for subsequent evaluation phases.

## A.6 Inference Results

### A.6.1 SL

We have conducted comprehensive testing of our SL model across various topographies and environments, examining the performance under different conditions. The overall performance comparison

**Algorithm 2** DVN with experience replay

1: **Input:** states per episode $n$, batch size $m$, action budget $k$, parameter update interval $T'$, max number of episodes $e_{\max}$
2: **Output:** V network $\hat{V}_\theta$
3: Initialize experience replay memory $D$
4: Initialize policy network $V$ with random weights $\theta$
5: Initialize target network $\hat{V}$ with weights $\theta^-$
6: Initialize $\epsilon$-decay to 1 and anneal to 0.1 with training
7: **for** $e = 1$ to $e_{\max}$ **do**
8:     $t \leftarrow 1$
9:     Generate $n$ random states $s_t$ with initial infected nodes
10:     Calculate candidate node set $C$ for $s_t$
11:     **while** any $|C| > 0$ **do**
12:         Initialize blocker set $B_t \leftarrow \emptyset$
13:         Randomly sample a number $x$ from uniform distribution $\mathcal{U}(0, 1)$
14:         **if** $x < \epsilon$ **then**
15:             Randomly sample $k$ candidates from $C$ as blocker set $B_t$
16:         **else**
17:             Initialize the infection prediction set $K \leftarrow \emptyset$
18:             **for** all node $u$ of candidate set $C$ **do**
19:                 Calculate the infection number $K_u$ using $V_\pi(s_{t+1})$, where $s_{t+1}$ is the state resulting after taking action $u$ in state $s_t$
20:                 Append $K_u$ to $K$
21:             **end for**
22:             Select the $k$ nodes with the least infection prediction from $K$ as the blocker set $B_t$
23:         **end if**
24:         Block the nodes in the blocker set $B_t$
25:         Update the state $s_{t+1}$
26:         Update the candidate set $C$
27:         Update the reward $r_t$
28:         $t \leftarrow t + 1$
29:         **for** $i = 0$ to $n - 1$ **do**
30:             Store the transition $(s_{t-1}^i, B_t^i, r_t^i, s_t^i)$ in $D$
31:         **end for**
32:         Sample a random minibatch of $m$ transitions $(s_j, a_j, r_j, s_{j+1})$ from $D$
33:         $y_j \leftarrow \begin{cases} r_j & \text{if episode terminates at step } j + 1 \\ r_j + \hat{V}_{\theta^-(s_{j+1})} & \text{otherwise} \end{cases}$
34:         Calculate loss using mean squared error between $y_j$ and $V_\theta(s_j)$, set gradients to zero, perform backpropagation, and update weights using the Adam optimizer
35:         **if** $t \bmod T' = 0$ **then**
36:             Update target network $\theta^- \leftarrow \theta$
37:         **end if**
38:     **end while**
39: **end for**

for each model under these varied conditions is illustrated in Figure 3. This figure provides a comprehensive view of how the model performs across the different environments and topographical scenarios.

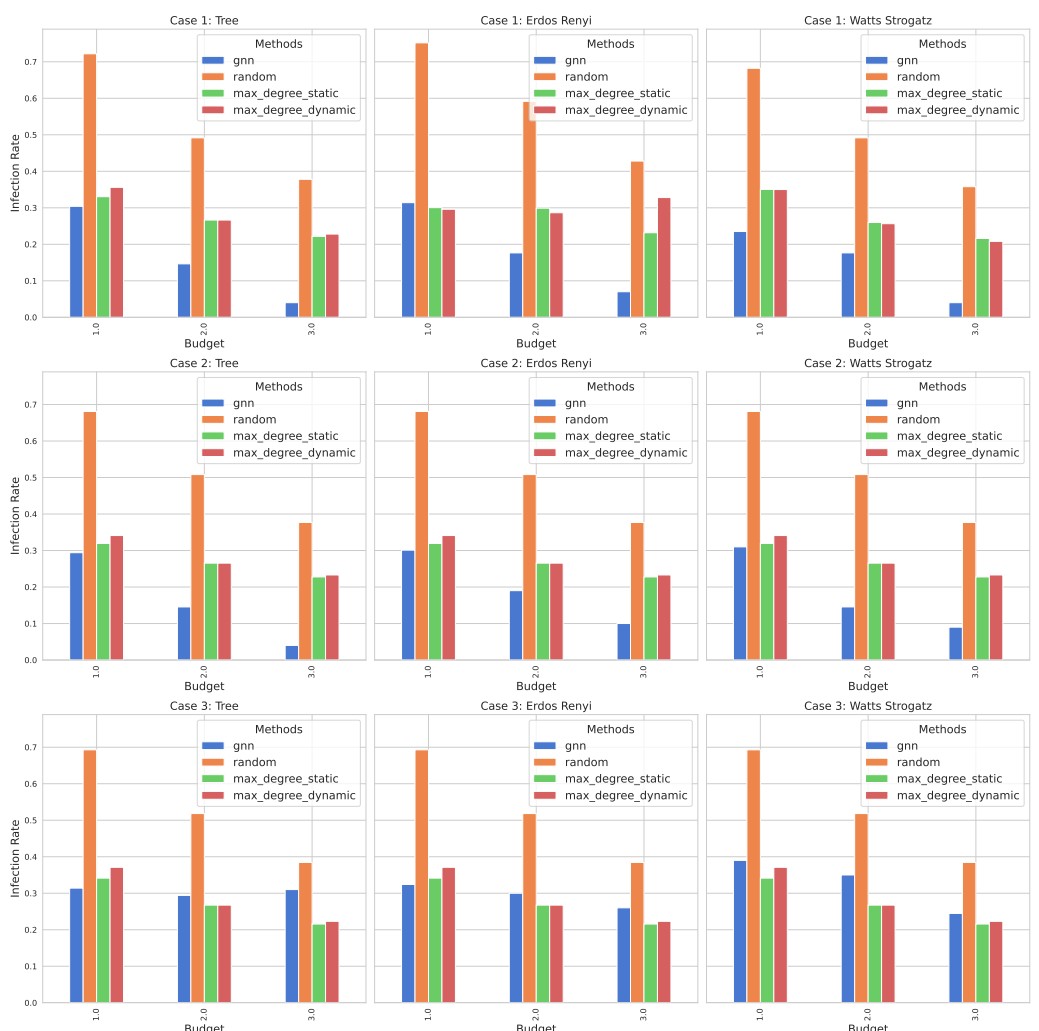

Figure 3: Comparative analysis of the GCN-Based SL Model Against Baseline Models Across Different Network Types and Budgets. Each subfigure represents one of the three cases (1, 2, and 3), organized by rows, for three different types of networks: Tree, Erdős-Rényi, and Watts-Strogatz, organized by columns. Within each panel, the infection rate is plotted for four methodologies. SL based on GCN (blue), random node selection (orange), static selection of maximum degrees (green), and dynamic selection of maximum degrees (red) across three levels of budget (1, 2, and 3). These results underscore the variability in performance with changes in network structure and budget allocation, highlighting the superior effectiveness of the GCN model in simpler cases and under increased budget conditions, with diminishing returns in more complex environments.

**Dataset v2 Testing** In addition to the initial dataset, we have also tested our model using dataset v2. For a more granular analysis, we have compiled the test results into three distinct cases:

- **Case 1:** Detailed in Figure 4

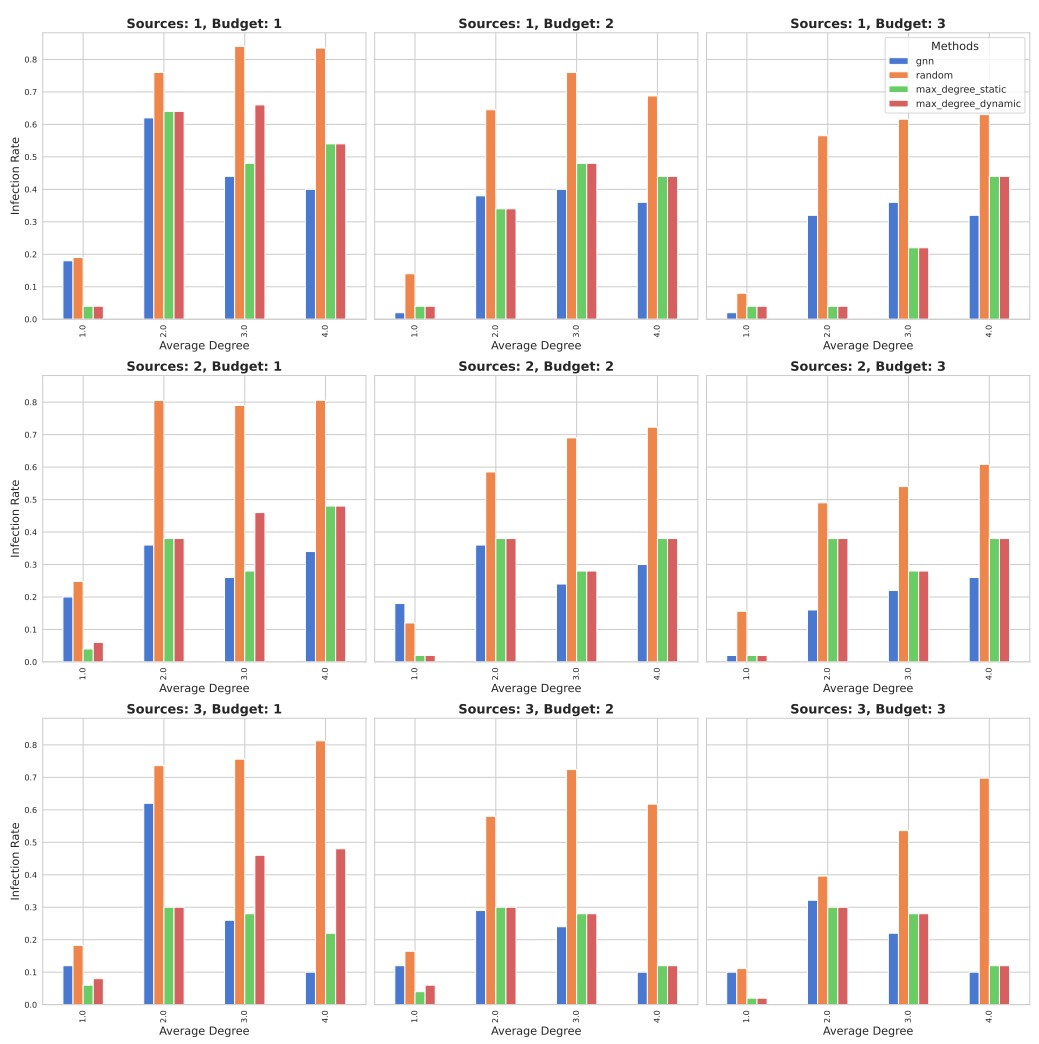

Figure 4: Case-1: Comparative Mean Infection Rate across different parameter settings for a GCN-based SL model trained on a 25-node dataset and tested on dataset v2 consisting of 50 nodes

- **Case 2:** Detailed in Figure 5

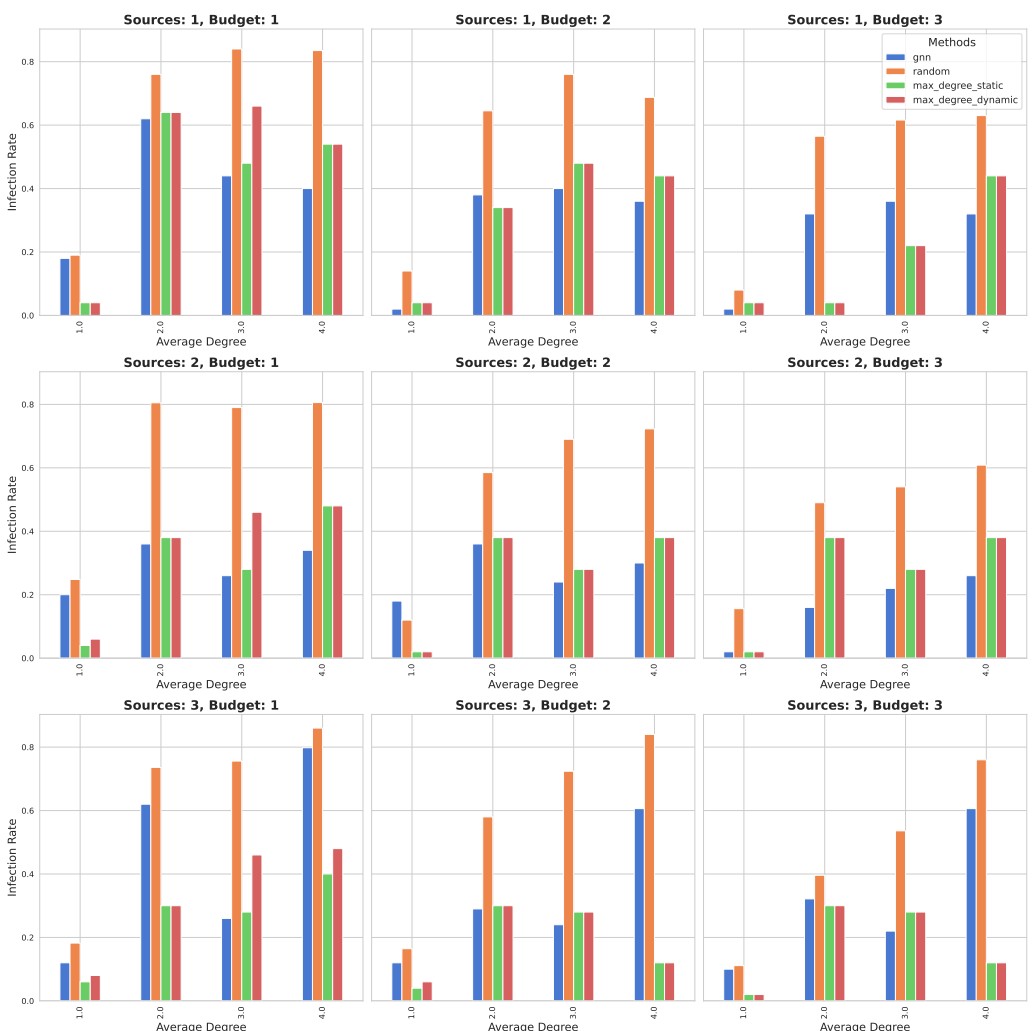

Figure 5: Case-2: Comparative Mean Infection Rate across different parameter settings for a GCN-based SL model trained on a 25-node dataset and tested on dataset v2 consisting of 50 nodes

- **Case 3:** Detailed in Figure 6

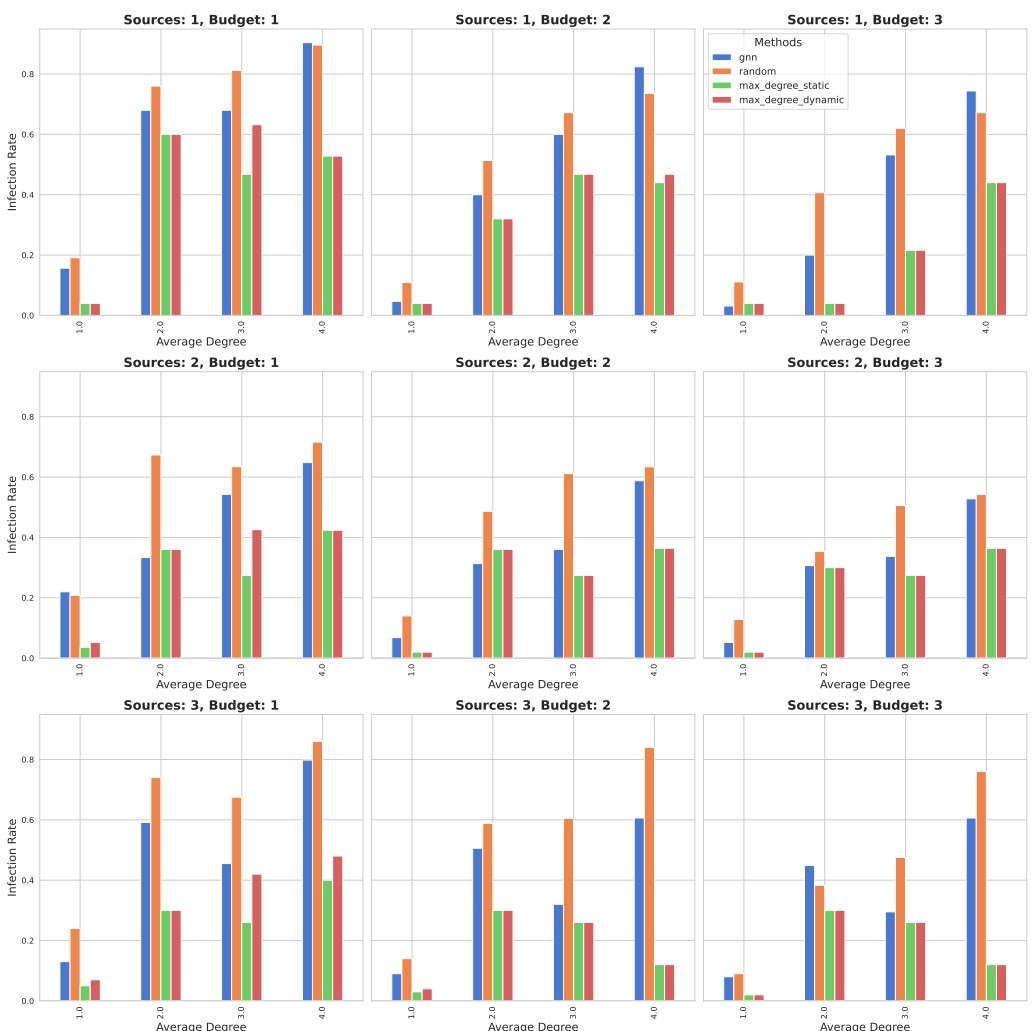

Figure 6: Case-3: Comparative Mean Infection Rate across different parameter settings for a GCN-based SL model trained on a 25-node dataset and tested on dataset v2 consisting of 50 nodes

### A.6.2 RL

Comparison of MSE loss across different reward functions for Case-1: Figure 7.

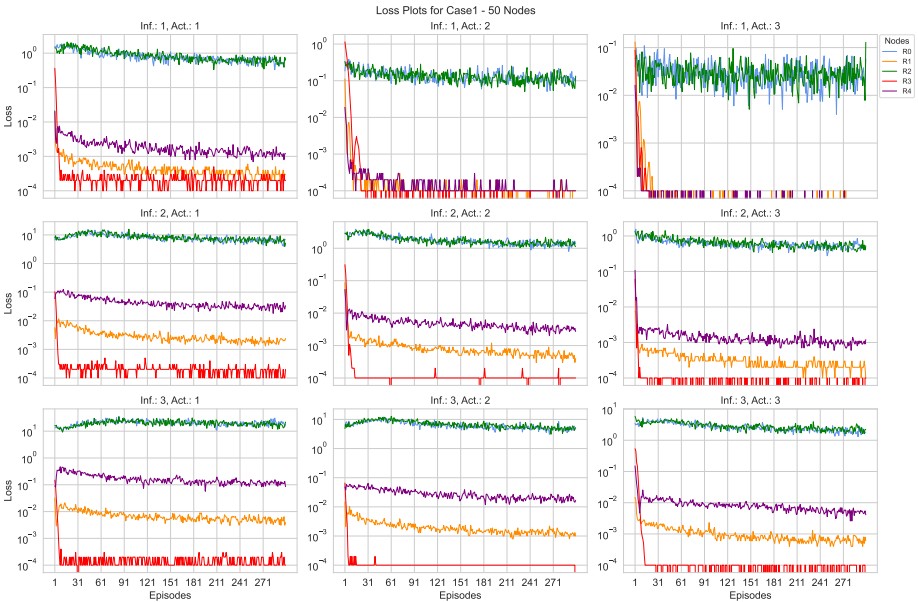

Figure 7: Case-1: Comparative MSE loss across different reward functions for a ResNet model trained on a 50-node dataset. Columns represent an increase in action budget during training, while rows indicate a rise in the number of initial infected nodes.

Comparison of Mean Infection Rate across different reward functions, showcasing that the reward $R_3$ performs better in model configurations with higher action budget and higher initial infected nodes: Figure 8

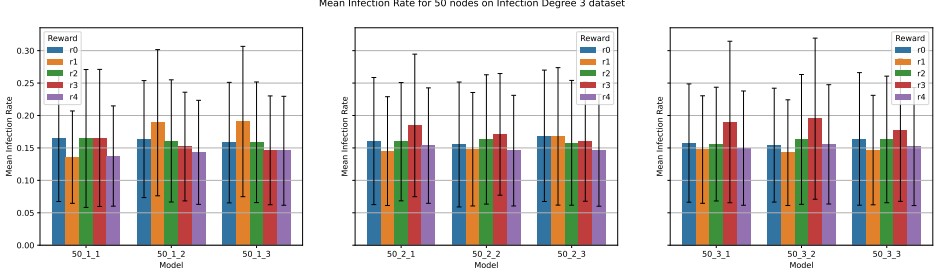

Figure 8: Case-1: Comparative Mean Infection Rate across different reward functions for a ResNet model trained on a 50-node dataset tested on Dataset v2 of 50 nodes with degree of connectivity 3.

Case-1

- **Type:** Binary Opinion and Binary Trust.
- **Opinion Dynamic Model:** Discrete Switching.

**R0** The loss plot is presented in Figure 9. Dataset v1 Inference Result: 50 Nodes - Figure 10.

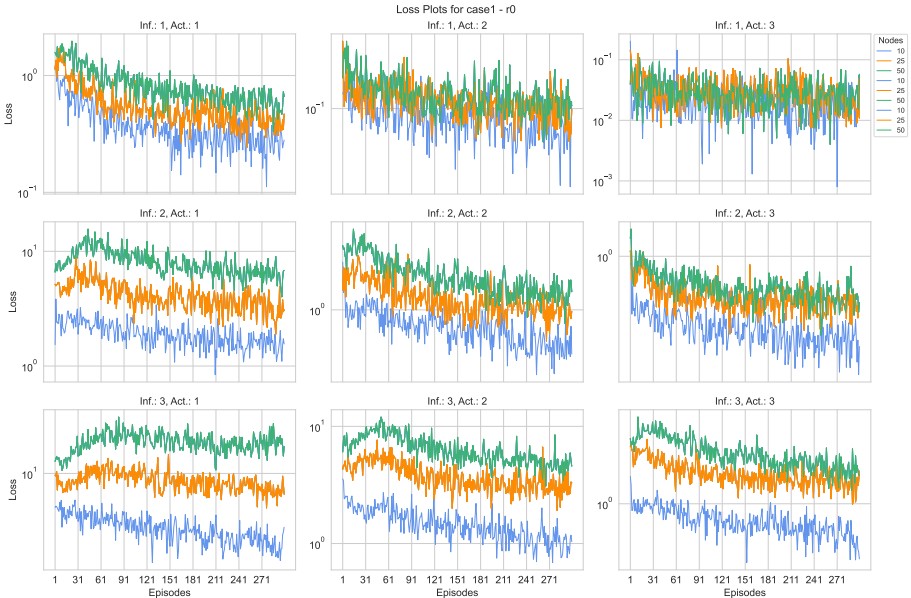

Figure 9: Case-1 using R0: Training MSE loss evolution for RL policy using ResNet model across networks of 10 (blue), 25 (orange), and 50 (green) nodes, for varying initial misinformation sources (Inf.) and action budgets (Act.). Plotted on a logarithmic scale, the loss decreases over episodes, indicating improved policy performance and adaptation across network sizes.

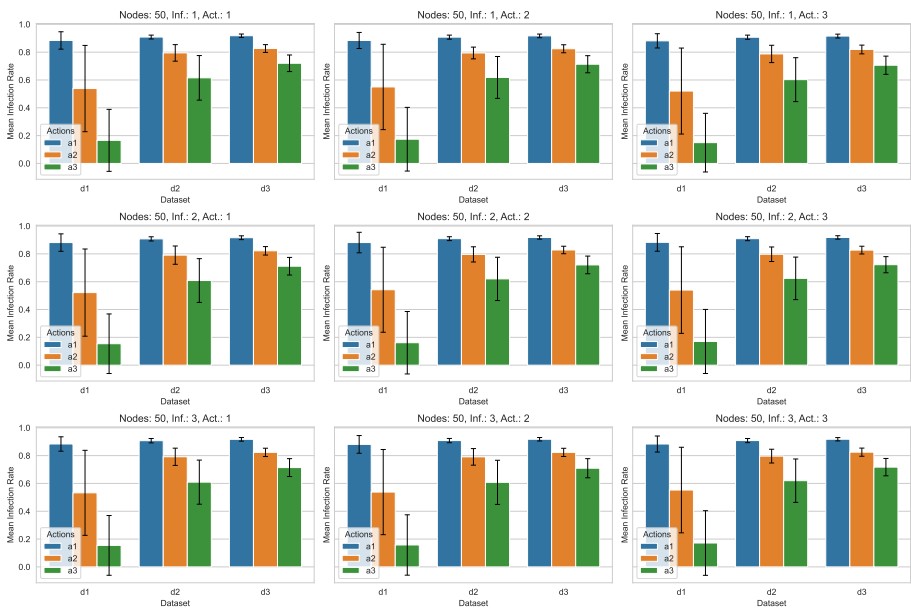

Figure 10: Case-1 using R0: Performance comparison of the RL policies trained using ResNet model for a 50-node network. The barplot displays the mean infection rate for datasets d1, d2, and d3 of Dataset v1 type, differentiated by the number of initial misinformation sources (Inf.) and action budgets (Act.: a1, a2, a3). Each subplot illustrates the performance of a policy trained with the parameters indicated in its title. Lower infection rates indicate more effective policy learning and misinformation containment.

**R1** The loss plot is presented in Figure 11. Dataset v1 Inference Result: 50 Nodes - Figure 12.

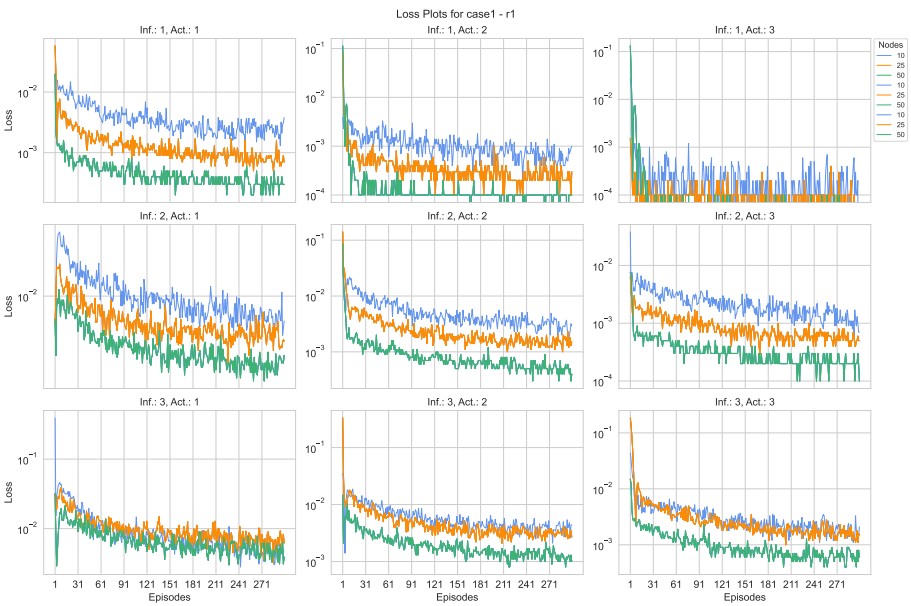

Figure 11: Case-1 using R1: Training MSE loss evolution for RL policy using ResNet model across networks of 10 (blue), 25 (orange), and 50 (green) nodes, for varying initial misinformation sources (Inf.) and action budgets (Act.). Plotted on a logarithmic scale, the loss decreases over episodes, indicating improved policy performance and adaptation across network sizes.

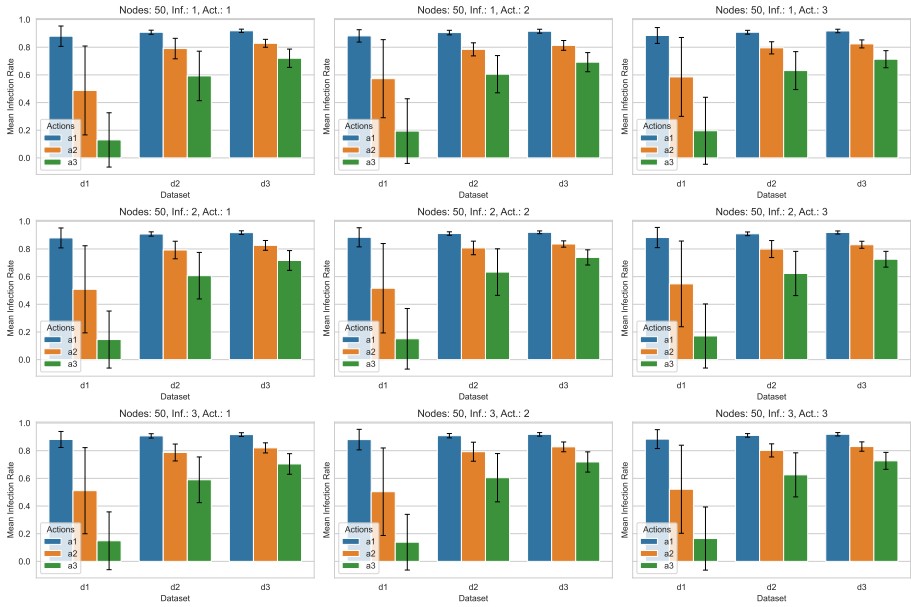

Figure 12: Case-1 using R1: Performance comparison of the RL policies trained using ResNet model for a 50-node network. The barplot displays the mean infection rate for datasets d1, d2, and d3 of Dataset v1 type, differentiated by the number of initial misinformation sources (Inf.) and action budgets (Act.: a1, a2, a3). Each subplot illustrates the performance of a policy trained with the parameters indicated in its title. Lower infection rates indicate more effective policy learning and misinformation containment.

**R2** The loss plot is presented in Figure 13. Dataset v1 Inference Result: 50 Nodes - Figure 14.

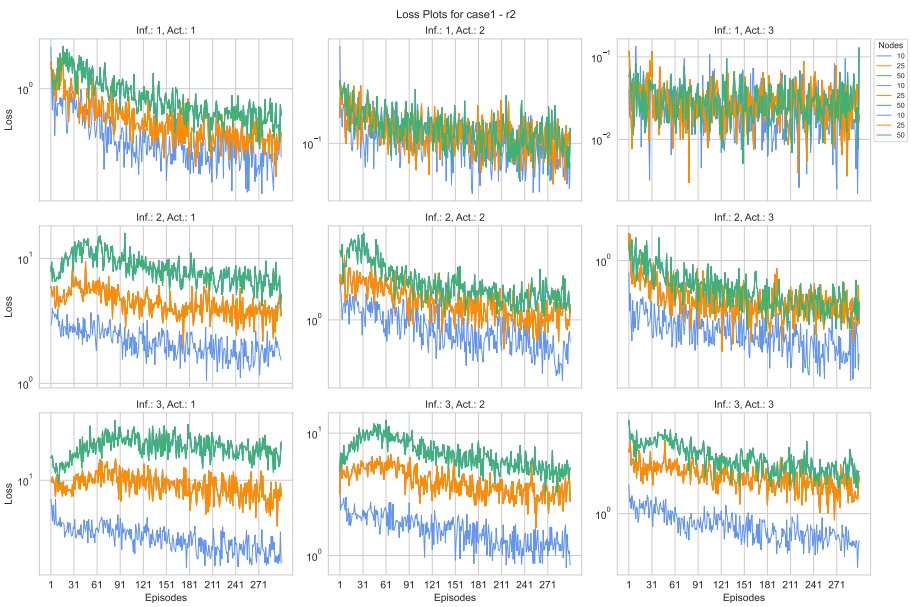

Figure 13: Case-1 using R2: Training MSE loss evolution for RL policy using ResNet model across networks of 10 (blue), 25 (orange), and 50 (green) nodes, for varying initial misinformation sources (Inf.) and action budgets (Act.). Plotted on a logarithmic scale, the loss decreases over episodes, indicating improved policy performance and adaptation across network sizes.

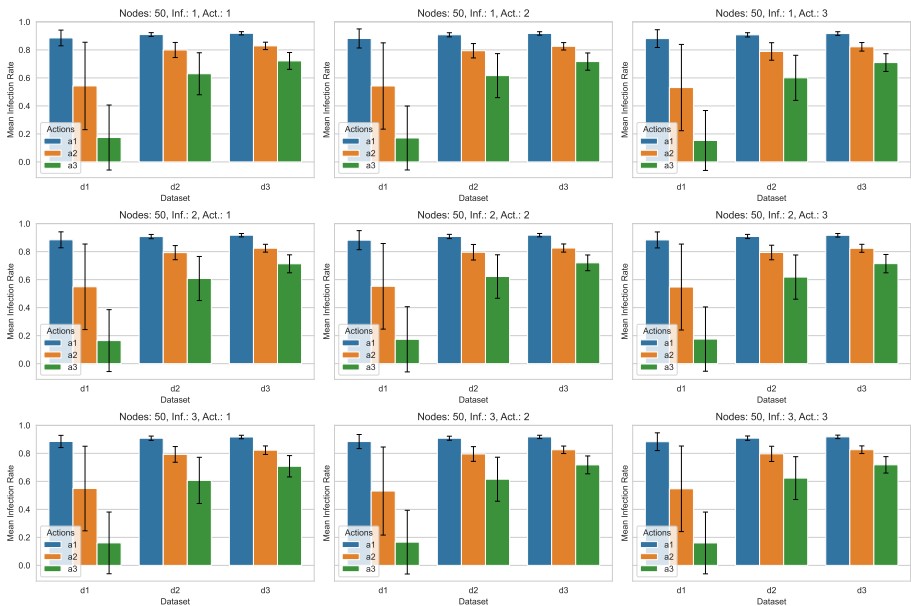

Figure 14: Case-1 using R2: Performance comparison of the RL policies trained using ResNet model for a 50-node network. The barplot displays the mean infection rate for datasets d1, d2, and d3 of Dataset v1 type, differentiated by the number of initial misinformation sources (Inf.) and action budgets (Act.: a1, a2, a3). Each subplot illustrates the performance of a policy trained with the parameters indicated in its title. Lower infection rates indicate more effective policy learning and misinformation containment.

**R3**    The loss plot is presented in Figure 15. Dataset v1 Inference Result: 50 Nodes - Figure 16.

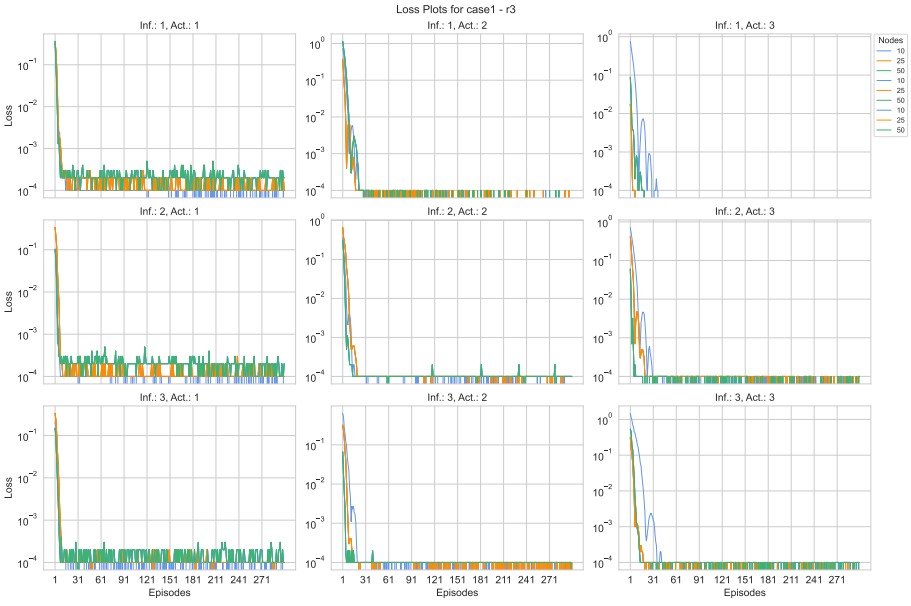

Figure 15: Case-1 using R3: Training MSE loss evolution for RL policy using ResNet model across networks of 10 (blue), 25 (orange), and 50 (green) nodes, for varying initial misinformation sources (Inf.) and action budgets (Act.). Plotted on a logarithmic scale, the loss decreases over episodes, indicating improved policy performance and adaptation across network sizes.

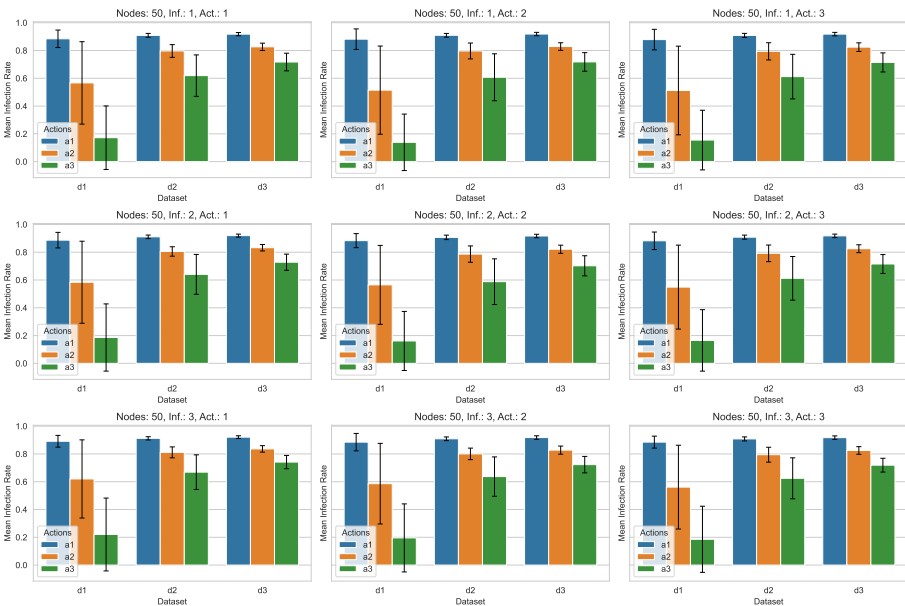

Figure 16: Case-1 using R3: Performance comparison of the RL policies trained using ResNet model for a 50-node network. The barplot displays the mean infection rate for datasets d1, d2, and d3 of Dataset v1 type, differentiated by the number of initial misinformation sources (Inf.) and action budgets (Act.: a1, a2, a3). Each subplot illustrates the performance of a policy trained with the parameters indicated in its title. Lower infection rates indicate more effective policy learning and misinformation containment.

**R4** The loss plot is presented in Figure 17. Dataset v1 Inference Result: 50 Nodes - Figure 18.

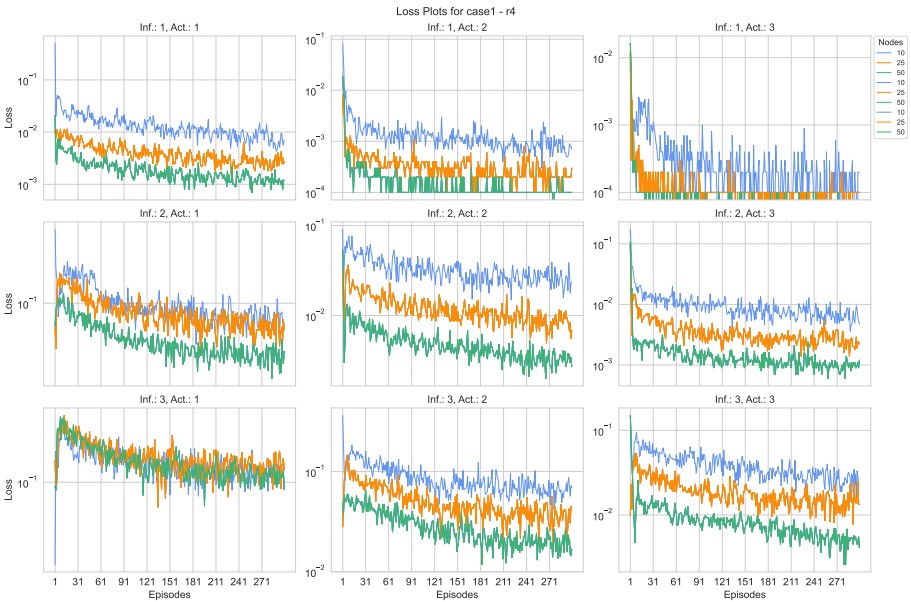

Figure 17: Case-1 using R4: Training MSE loss evolution for RL policy using ResNet model across networks of 10 (blue), 25 (orange), and 50 (green) nodes, for varying initial misinformation sources (Inf.) and action budgets (Act.). Plotted on a logarithmic scale, the loss decreases over episodes, indicating improved policy performance and adaptation across network sizes.

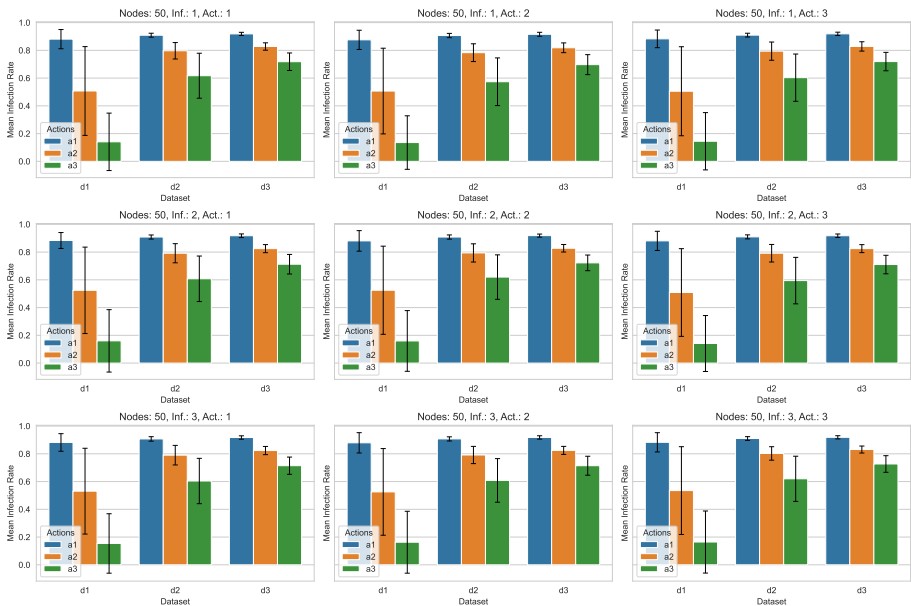

Figure 18: Case-1 using R4: Performance comparison of the RL policies trained using ResNet model for a 50-node network. The barplot displays the mean infection rate for datasets d1, d2, and d3 of Dataset v1 type, differentiated by the number of initial misinformation sources (Inf.) and action budgets (Act.: a1, a2, a3). Each subplot illustrates the performance of a policy trained with the parameters indicated in its title. Lower infection rates indicate more effective policy learning and misinformation containment.

**Dataset v2 Results**

**Degree of connectivity 1**     50 Nodes - Figure 19

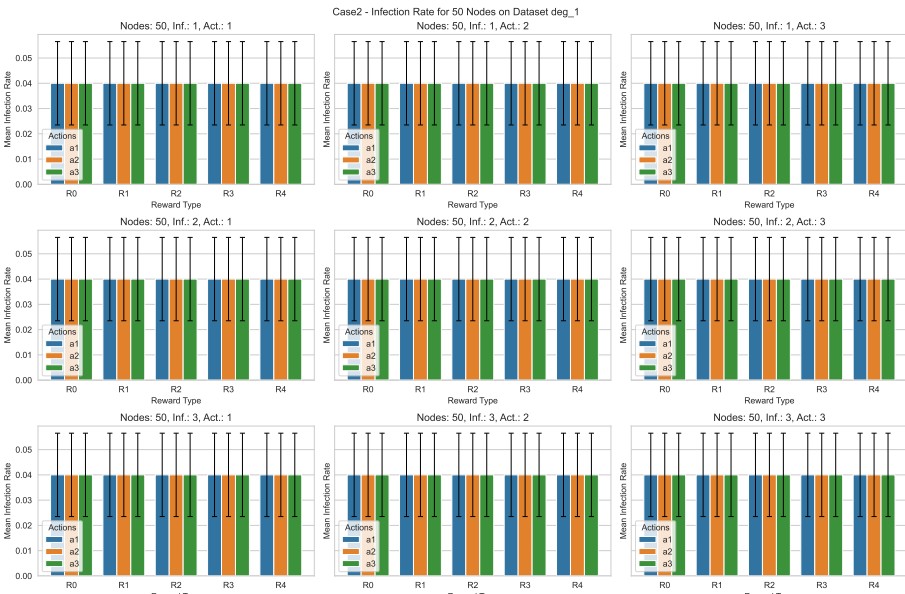

Figure 19: Case-1 inference on Dataset v2: Performance comparison of the RL policies trained using ResNet model for a 50-node network. The barplot displays the mean infection rate for different reward functions on Dataset v2 of Degree of connectivity 1, differentiated by the number of initial misinformation sources (Inf.) and action budgets (Act.: a1, a2, a3). Each subplot illustrates the performance of a policy trained with the parameters indicated in its title. Lower infection rates indicate more effective policy learning and misinformation containment.

**Degree of connectivity 2** 50 Nodes - Figure 20

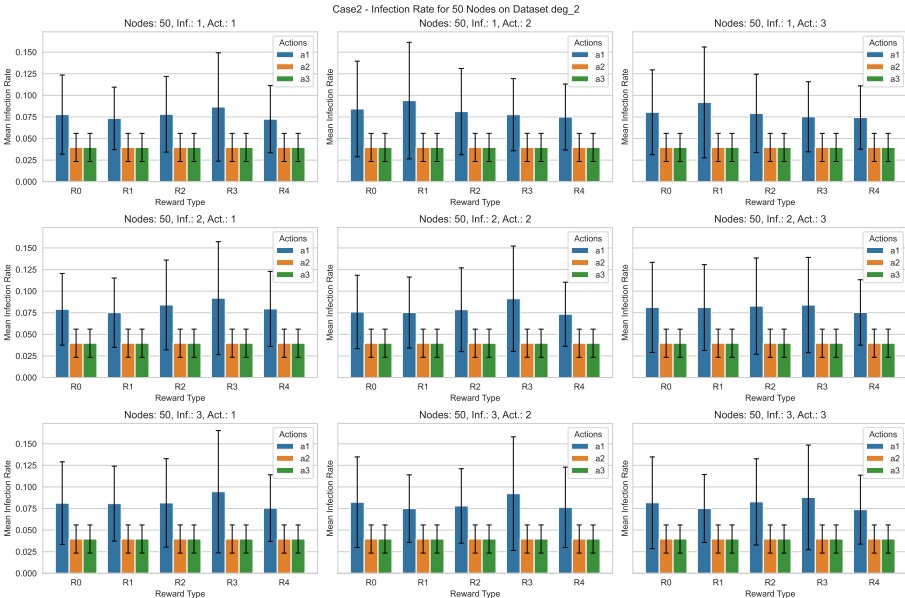

Figure 20: Case-1 inference on Dataset v2: Performance comparison of the RL policies trained using ResNet model for a 50-node network. The barplot displays the mean infection rate for different reward functions on Dataset v2 of Degree of connectivity 2, differentiated by the number of initial misinformation sources (Inf.) and action budgets (Act.: a1, a2, a3). Each subplot illustrates the performance of a policy trained with the parameters indicated in its title. Lower infection rates indicate more effective policy learning and misinformation containment.

**Degree of connectivity 3** 50 Nodes - Figure 21

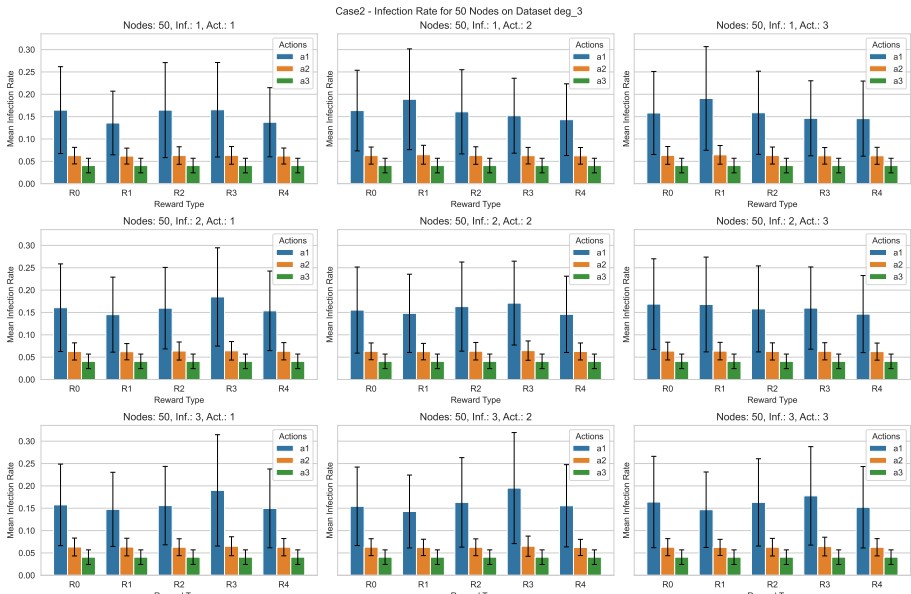

Figure 21: Case-1 inference on Dataset v2: Performance comparison of the RL policies trained using ResNet model for a 50-node network. The barplot displays the mean infection rate for different reward functions on Dataset v2 of Degree of connectivity 3, differentiated by the number of initial misinformation sources (Inf.) and action budgets (Act.: a1, a2, a3). Each subplot illustrates the performance of a policy trained with the parameters indicated in its title. Lower infection rates indicate more effective policy learning and misinformation containment.

**Degree of connectivity 4**    50 Nodes - Figure 22

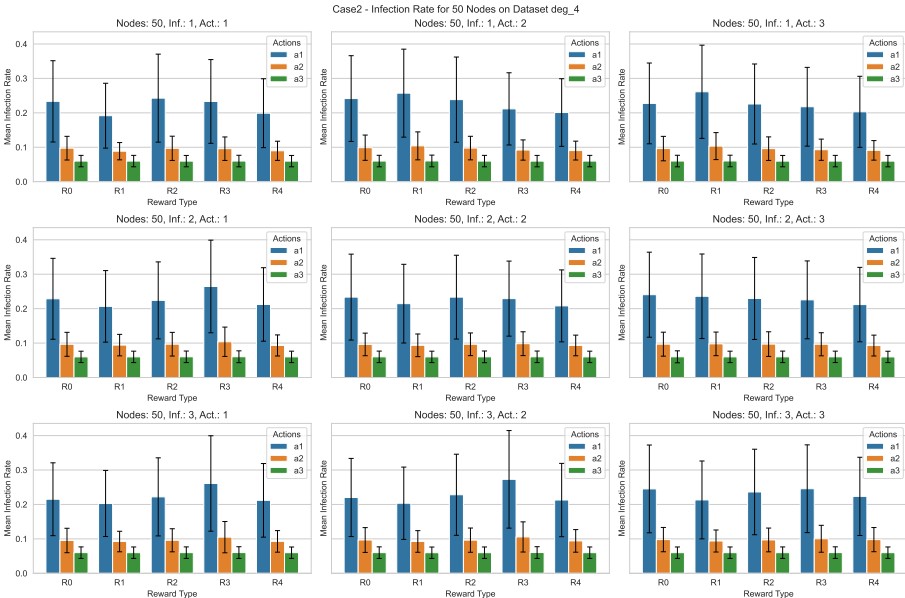

Figure 22: Case-1 inference on Dataset v2: Performance comparison of the RL policies trained using ResNet model for a 50-node network. The barplot displays the mean infection rate for different reward functions on Dataset v2 of Degree of connectivity 4, differentiated by the number of initial misinformation sources (Inf.) and action budgets (Act.: a1, a2, a3). Each subplot illustrates the performance of a policy trained with the parameters indicated in its title. Lower infection rates indicate more effective policy learning and misinformation containment.

**Case-2**

- **Type:** Floating Point Opinion and Binary Trust.
- **Opinion Dynamic Model:** Linear Adjustment.

**R0** The loss plot is presented in Figure 23. Dataset v1 Inference Result: 50 Nodes - Figure 24.

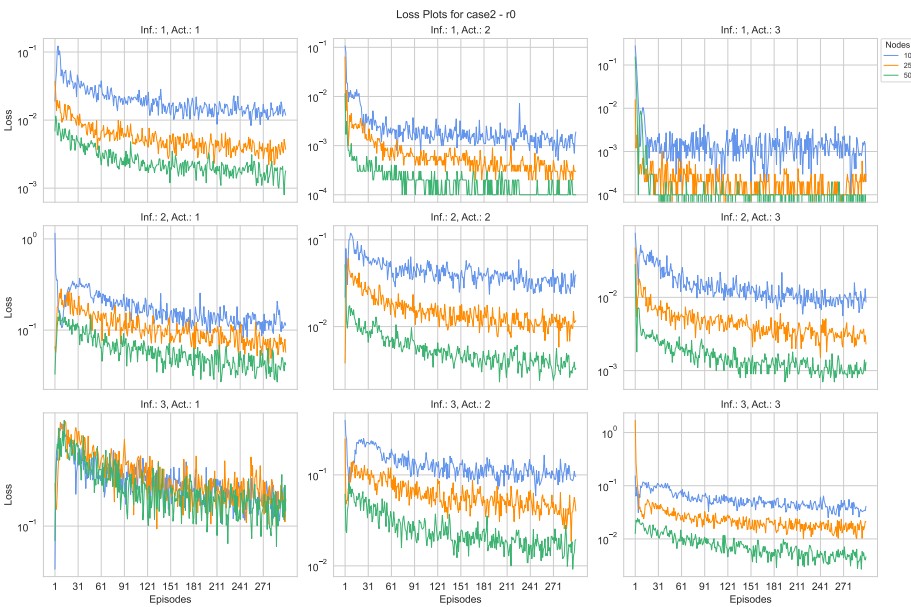

Figure 23: Case-2 using R0: Training MSE loss evolution for RL policy using ResNet model across networks of 10 (blue), 25 (orange), and 50 (green) nodes, for varying initial misinformation sources (Inf.) and action budgets (Act.). Plotted on a logarithmic scale, the loss decreases over episodes, indicating improved policy performance and adaptation across network sizes.

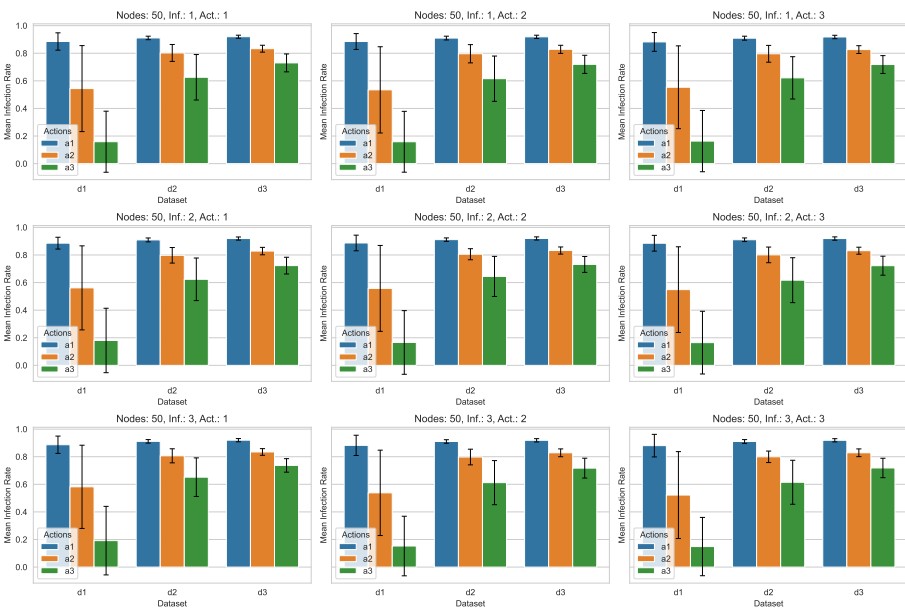

Figure 24: Case-2 using R0: Performance comparison of the RL policies trained using ResNet model for a 50-node network. The barplot displays the mean infection rate for datasets d1, d2, and d3 of Dataset v1 type, differentiated by the number of initial misinformation sources (Inf.) and action budgets (Act.: a1, a2, a3). Each subplot illustrates the performance of a policy trained with the parameters indicated in its title. Lower infection rates indicate more effective policy learning and misinformation containment.

**R1** The loss plot is presented in Figure 25. Dataset v1 Inference Result: 50 Nodes - Figure 26.

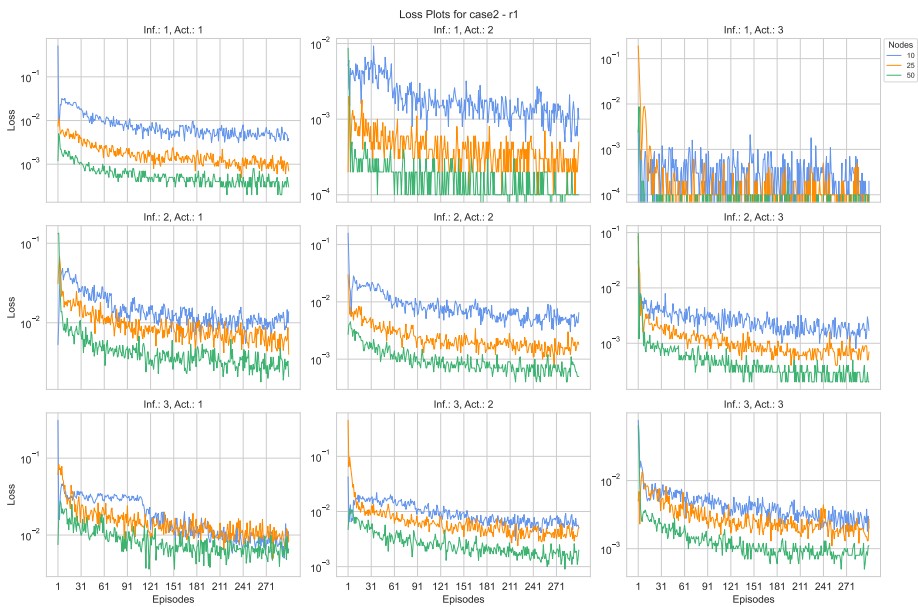

Figure 25: Case-2 using R1: Training MSE loss evolution for RL policy using ResNet model across networks of 10 (blue), 25 (orange), and 50 (green) nodes, for varying initial misinformation sources (Inf.) and action budgets (Act.). Plotted on a logarithmic scale, the loss decreases over episodes, indicating improved policy performance and adaptation across network sizes.

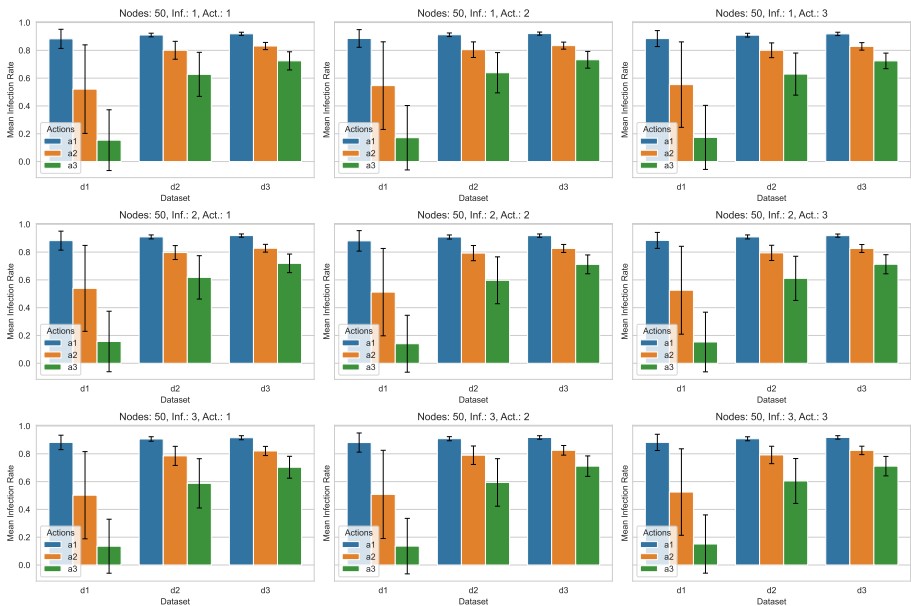

Figure 26: Case-2 using R1: Performance comparison of the RL policies trained using ResNet model for a 50-node network. The barplot displays the mean infection rate for datasets d1, d2, and d3 of Dataset v1 type, differentiated by the number of initial misinformation sources (Inf.) and action budgets (Act.: a1, a2, a3). Each subplot illustrates the performance of a policy trained with the parameters indicated in its title. Lower infection rates indicate more effective policy learning and misinformation containment.

**R2** The loss plot is presented in Figure 27. Dataset v1 Inference Result: 50 Nodes - Figure 28.

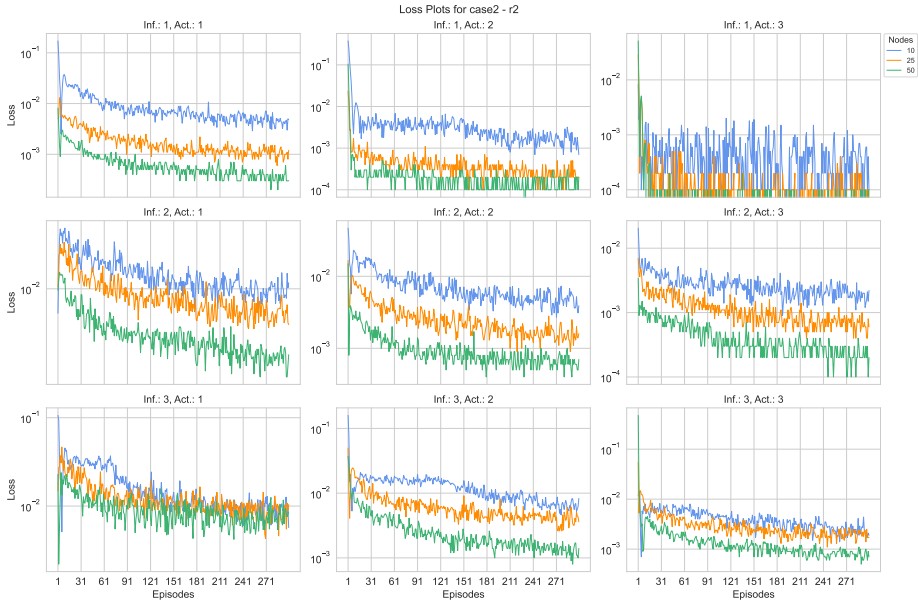

Figure 27: Case-2 using R2: Training MSE loss evolution for RL policy using ResNet model across networks of 10 (blue), 25 (orange), and 50 (green) nodes, for varying initial misinformation sources (Inf.) and action budgets (Act.). Plotted on a logarithmic scale, the loss decreases over episodes, indicating improved policy performance and adaptation across network sizes.

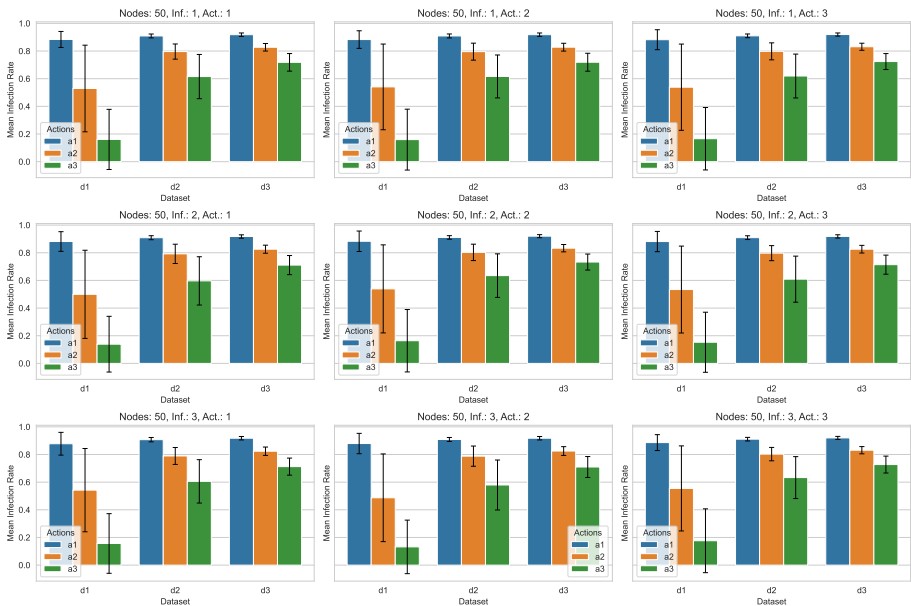

Figure 28: Case-2 using R2: Performance comparison of the RL policies trained using ResNet model for a 50-node network. The barplot displays the mean infection rate for datasets d1, d2, and d3 of Dataset v1 type, differentiated by the number of initial misinformation sources (Inf.) and action budgets (Act.: a1, a2, a3). Each subplot illustrates the performance of a policy trained with the parameters indicated in its title. Lower infection rates indicate more effective policy learning and misinformation containment.

**R3** The loss plot is presented in Figure 29. Dataset v1 Inference Result: 50 Nodes - Figure 30.

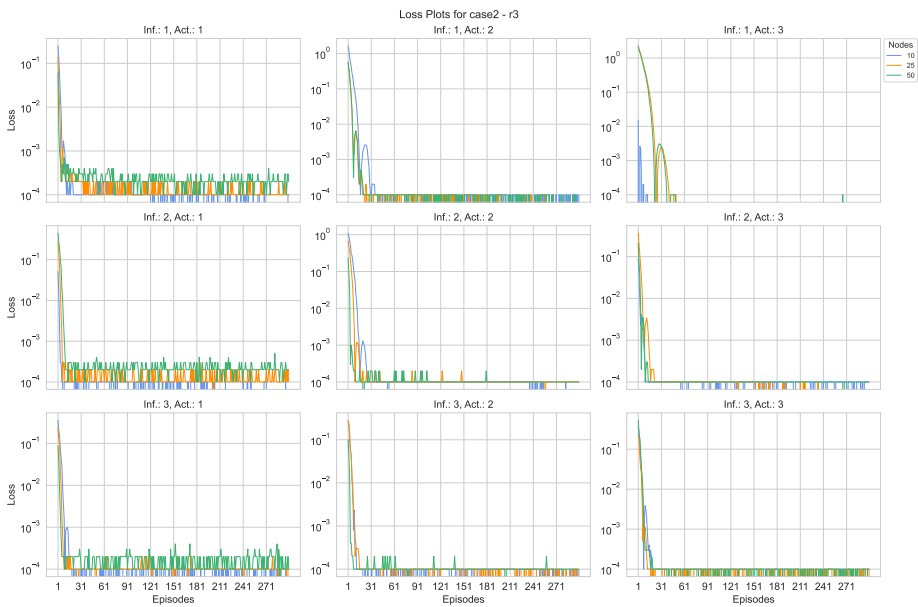

Figure 29: Case-2 using R3: Training MSE loss evolution for RL policy using ResNet model across networks of 10 (blue), 25 (orange), and 50 (green) nodes, for varying initial misinformation sources (Inf.) and action budgets (Act.). Plotted on a logarithmic scale, the loss decreases over episodes, indicating improved policy performance and adaptation across network sizes.

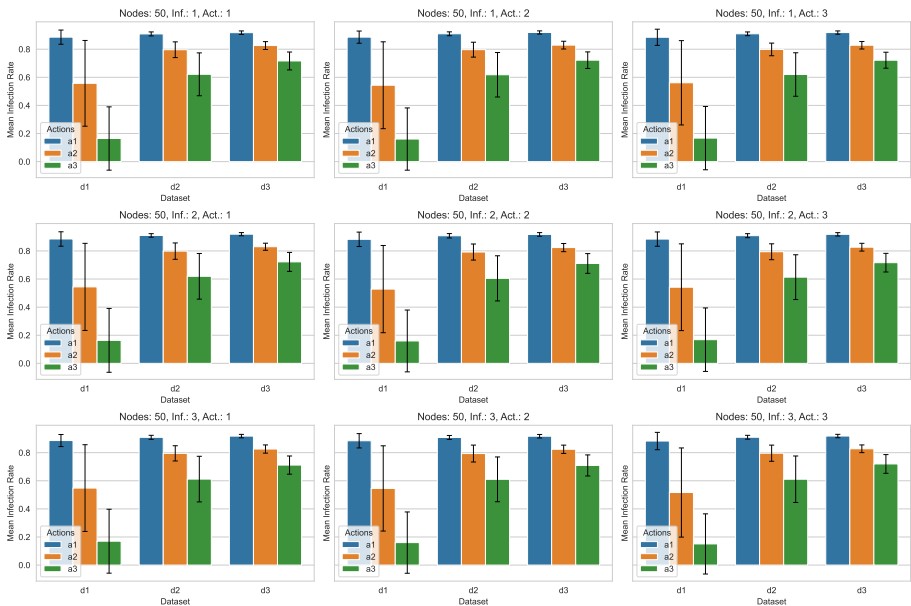

Figure 30: Case-2 using R3: Performance comparison of the RL policies trained using ResNet model for a 50-node network. The barplot displays the mean infection rate for datasets d1, d2, and d3 of Dataset v1 type, differentiated by the number of initial misinformation sources (Inf.) and action budgets (Act.: a1, a2, a3). Each subplot illustrates the performance of a policy trained with the parameters indicated in its title. Lower infection rates indicate more effective policy learning and misinformation containment.

**R4** The loss plot is presented in Figure 31. Dataset v1 Inference Result: 50 Nodes - Figure 32.

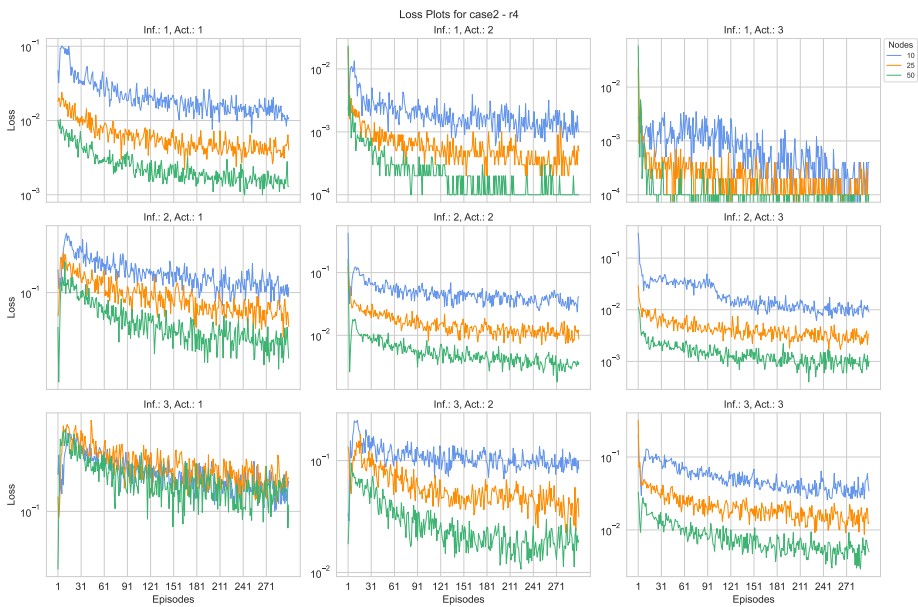

Figure 31: Case-2 using R4: Training MSE loss evolution for RL policy using ResNet model across networks of 10 (blue), 25 (orange), and 50 (green) nodes, for varying initial misinformation sources (Inf.) and action budgets (Act.). Plotted on a logarithmic scale, the loss decreases over episodes, indicating improved policy performance and adaptation across network sizes.

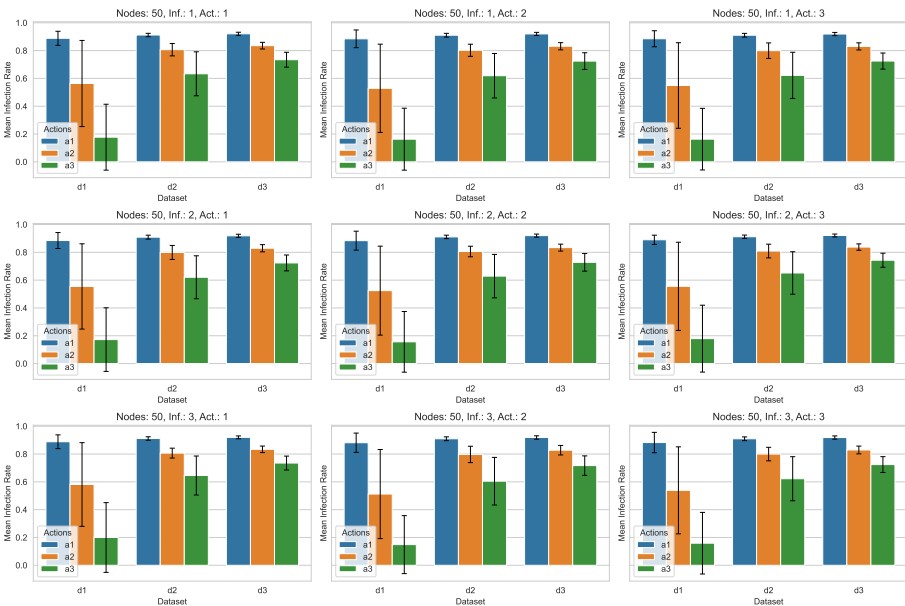

Figure 32: Case-2 using R4: Performance comparison of the RL policies trained using ResNet model for a 50-node network. The barplot displays the mean infection rate for datasets d1, d2, and d3 of Dataset v1 type, differentiated by the number of initial misinformation sources (Inf.) and action budgets (Act.: a1, a2, a3). Each subplot illustrates the performance of a policy trained with the parameters indicated in its title. Lower infection rates indicate more effective policy learning and misinformation containment.

**Dataset v2 Results**

**Degree of connectivity 1**    50 Nodes - Figure 33

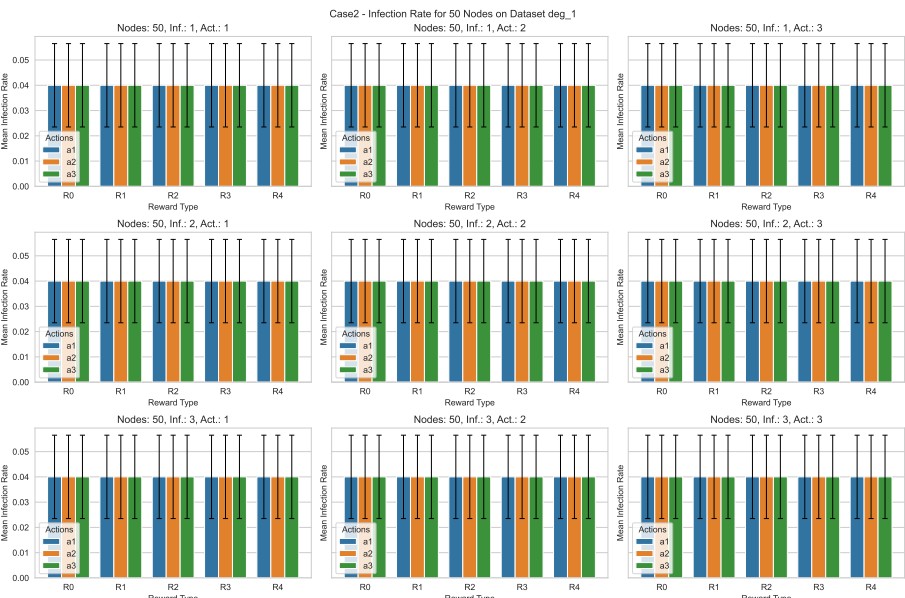

Figure 33: Case-2 inference on Dataset v2: Performance comparison of the RL policies trained using ResNet model for a 50-node network. The barplot displays the mean infection rate for different reward functions on Dataset v2 of Degree of Connectivity 1, differentiated by the number of initial misinformation sources (Inf.) and action budgets (Act.: a1, a2, a3). Each subplot illustrates the performance of a policy trained with the parameters indicated in its title. Lower infection rates indicate more effective policy learning and misinformation containment.

**Degree of connectivity 2**    50 Nodes - Figure 34

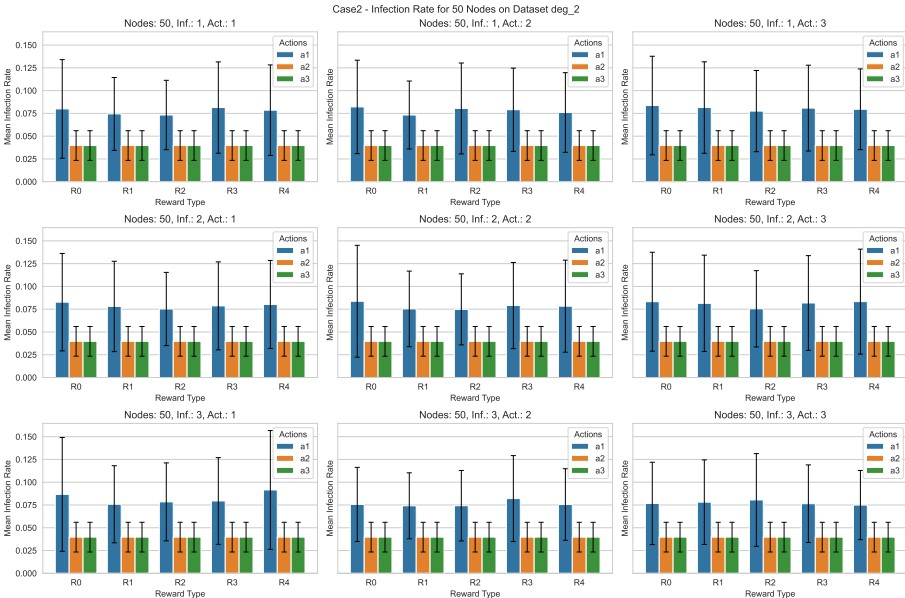

Figure 34: Case-2 inference on Dataset v2: Performance comparison of the RL policies trained using ResNet model for a 50-node network. The barplot displays the mean infection rate for different reward functions on Dataset v2 of Degree of Connectivity 2, differentiated by the number of initial misinformation sources (Inf.) and action budgets (Act.: a1, a2, a3). Each subplot illustrates the performance of a policy trained with the parameters indicated in its title. Lower infection rates indicate more effective policy learning and misinformation containment.

**Degree of connectivity 3**    50 Nodes - Figure 35

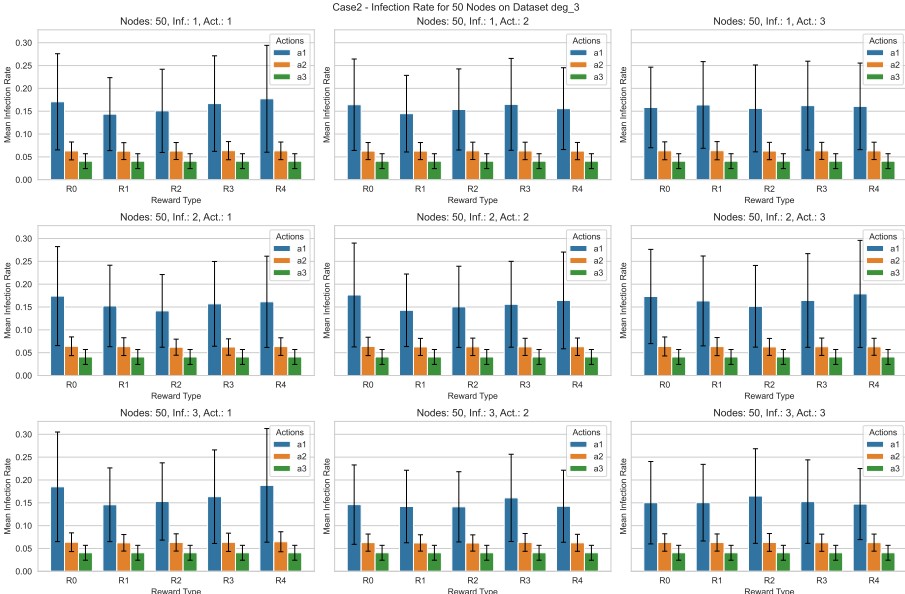

Figure 35: Case-2 inference on Dataset v2: Performance comparison of the RL policies trained using ResNet model for a 50-node network. The barplot displays the mean infection rate for different reward functions on Dataset v2 of Degree of Connectivity 3, differentiated by the number of initial misinformation sources (Inf.) and action budgets (Act.: a1, a2, a3). Each subplot illustrates the performance of a policy trained with the parameters indicated in its title. Lower infection rates indicate more effective policy learning and misinformation containment.

**Degree of connectivity 4**    50 Nodes - Figure 36

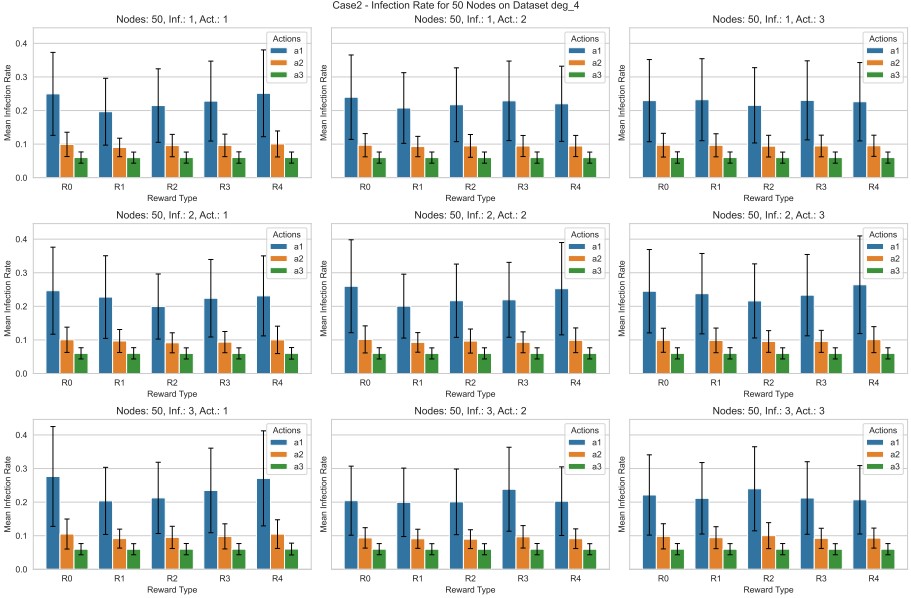

Figure 36: Case-2 inference on Dataset v2: Performance comparison of the RL policies trained using ResNet model for a 50-node network. The barplot displays the mean infection rate for different reward functions on Dataset v2 of Degree of Connectivity 4, differentiated by the number of initial misinformation sources (Inf.) and action budgets (Act.: a1, a2, a3). Each subplot illustrates the performance of a policy trained with the parameters indicated in its title. Lower infection rates indicate more effective policy learning and misinformation containment.

**Case-3**

**v1**

- **Type:** Floating Point Opinion and Floating Point Trust.
- **Opinion Dynamic Model:** Linear Adjustment.

**R0** The loss plot is presented in Figure 37. Dataset v1 Inference Result: 50 Nodes - Figure 38.

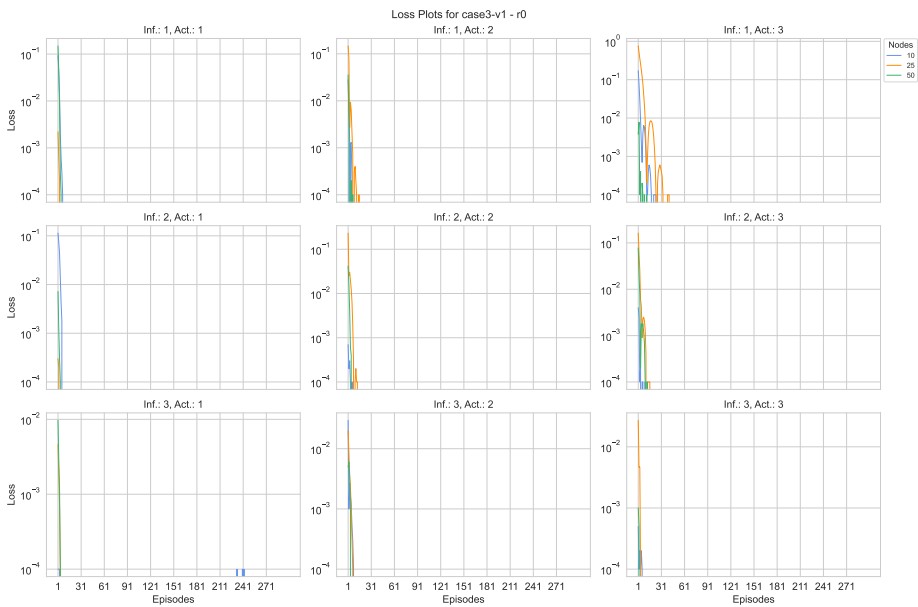

Figure 37: Case-3 v1 using R0: Training MSE loss evolution for RL policy using ResNet model across networks of 10 (blue), 25 (orange), and 50 (green) nodes, for varying initial misinformation sources (Inf.) and action budgets (Act.). The loss decreases over episodes, indicating improved policy performance and adaptation across network sizes.

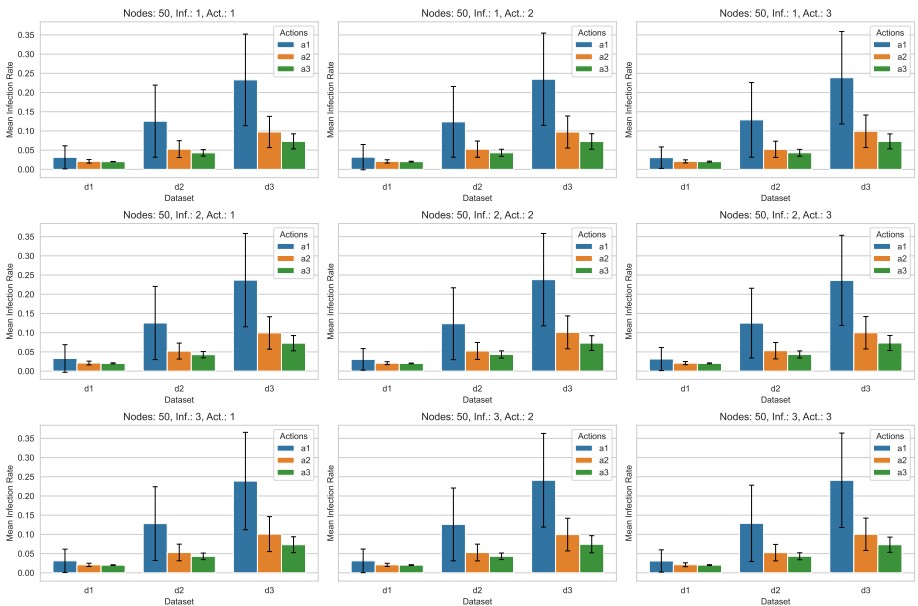

Figure 38: Case-3 v1 using R0: Performance comparison of the RL policies trained using ResNet model for a 50-node network. The barplot displays the mean infection rate for datasets d1, d2, and d3 of Dataset v1 type, differentiated by the number of initial misinformation sources (Inf.) and action budgets (Act.: a1, a2, a3). Each subplot illustrates the performance of a policy trained with the parameters indicated in its title. Lower infection rates indicate more effective policy learning and misinformation containment.

**R1** The loss plot is presented in Figure 39. Dataset v1 Inference Result: 50 Nodes - Figure 40.

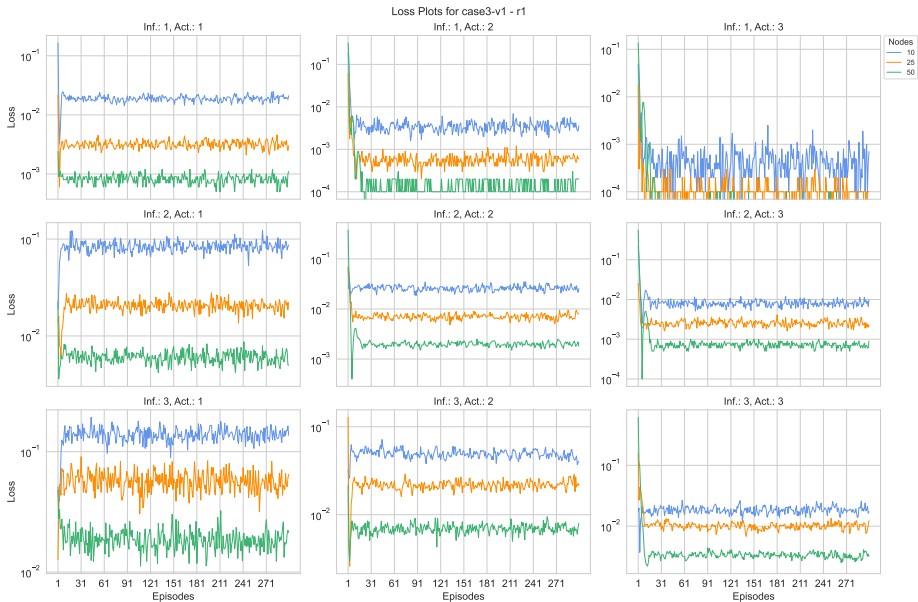

Figure 39: Case-3 v1 using R1: Training MSE loss evolution for RL policy using ResNet model across networks of 10 (blue), 25 (orange), and 50 (green) nodes, for varying initial misinformation sources (Inf.) and action budgets (Act.). Plotted on a logarithmic scale, the loss decreases over episodes, indicating improved policy performance and adaptation across network sizes.

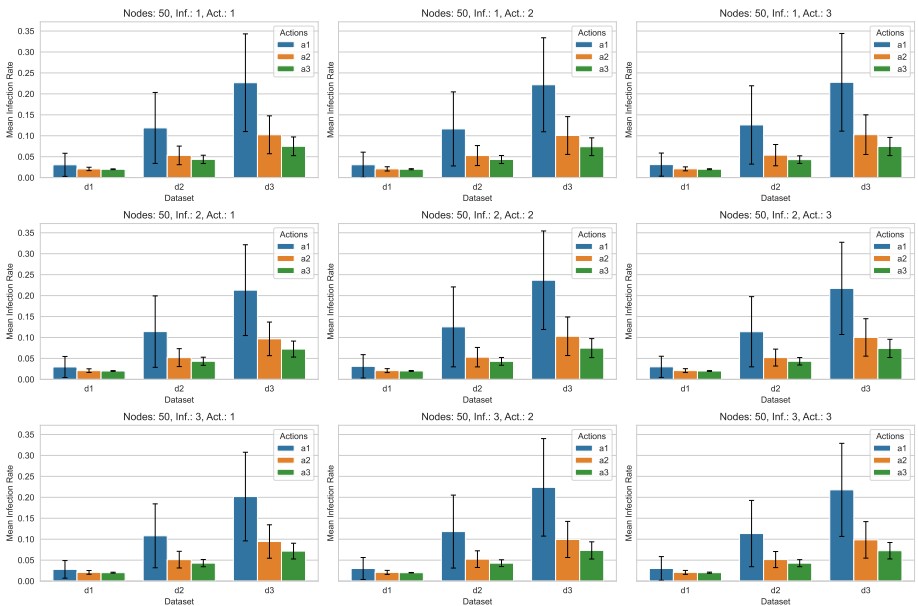

Figure 40: Case-3 v1 using R1: Performance comparison of the RL policies trained using ResNet model for a 50-node network. The barplot displays the mean infection rate for datasets d1, d2, and d3 of Dataset v1 type, differentiated by the number of initial misinformation sources (Inf.) and action budgets (Act.: a1, a2, a3). Each subplot illustrates the performance of a policy trained with the parameters indicated in its title. Lower infection rates indicate more effective policy learning and misinformation containment.

**R2** The loss plot is presented in Figure 41. Dataset v1 Inference Result: 50 Nodes - Figure 42.

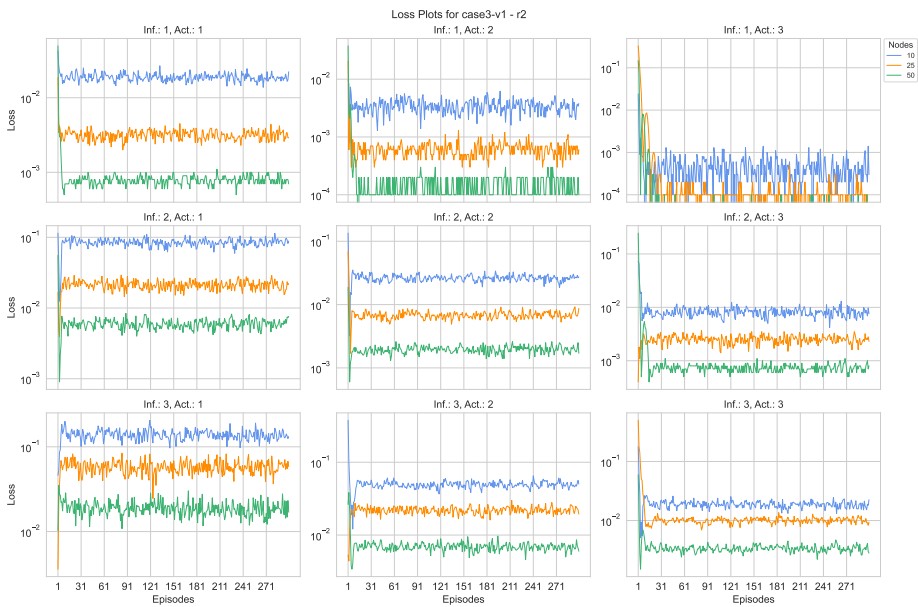

Figure 41: Case-3 v1 using R2: Training MSE loss evolution for RL policy using ResNet model across networks of 10 (blue), 25 (orange), and 50 (green) nodes, for varying initial misinformation sources (Inf.) and action budgets (Act.). Plotted on a logarithmic scale, the loss decreases over episodes, indicating improved policy performance and adaptation across network sizes.

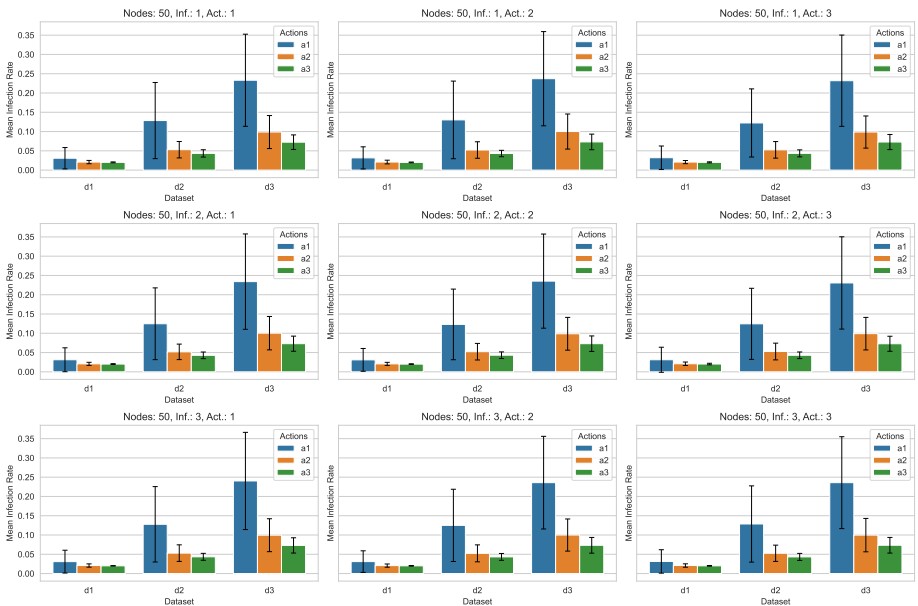

Figure 42: Case-3 v1 using R2: Performance comparison of the RL policies trained using ResNet model for a 50-node network. The barplot displays the mean infection rate for datasets d1, d2, and d3 of Dataset v1 type, differentiated by the number of initial misinformation sources (Inf.) and action budgets (Act.: a1, a2, a3). Each subplot illustrates the performance of a policy trained with the parameters indicated in its title. Lower infection rates indicate more effective policy learning and misinformation containment.

**R3**   The loss plot is presented in Figure 43. Dataset v1 Inference Result: 50 Nodes - Figure 44.

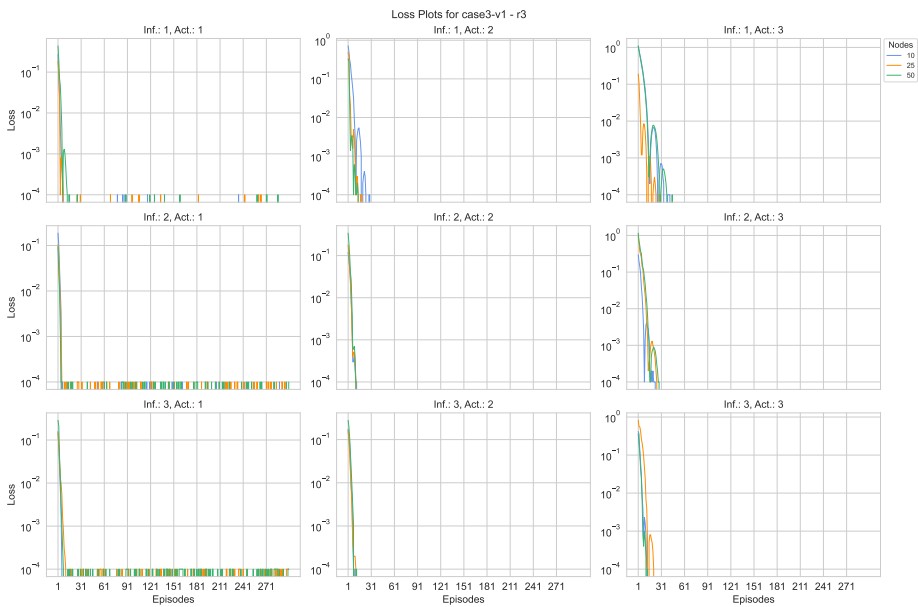

Figure 43: Case-3 v1 using R3: Training MSE loss evolution for RL policy using ResNet model across networks of 10 (blue), 25 (orange), and 50 (green) nodes, for varying initial misinformation sources (Inf.) and action budgets (Act.). The loss decreases over episodes, indicating improved policy performance and adaptation across network sizes.

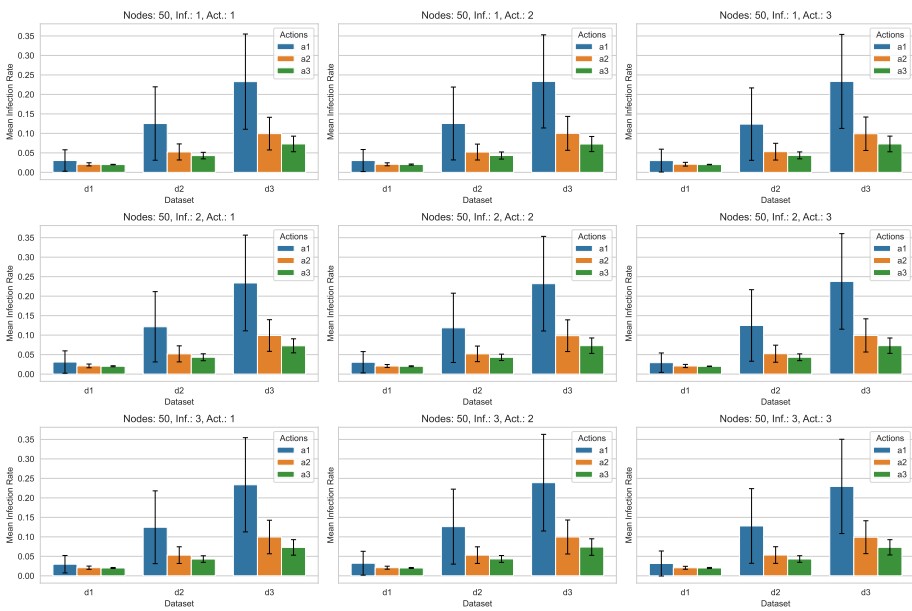

Figure 44: Case-3 v1 using R3: Performance comparison of the RL policies trained using ResNet model for a 50-node network. The barplot displays the mean infection rate for datasets d1, d2, and d3 of Dataset v1 type, differentiated by the number of initial misinformation sources (Inf.) and action budgets (Act.: a1, a2, a3). Each subplot illustrates the performance of a policy trained with the parameters indicated in its title. Lower infection rates indicate more effective policy learning and misinformation containment.

**R4** The loss plot is presented in Figure 45. Dataset v1 Inference Result: 50 Nodes - Figure 46.

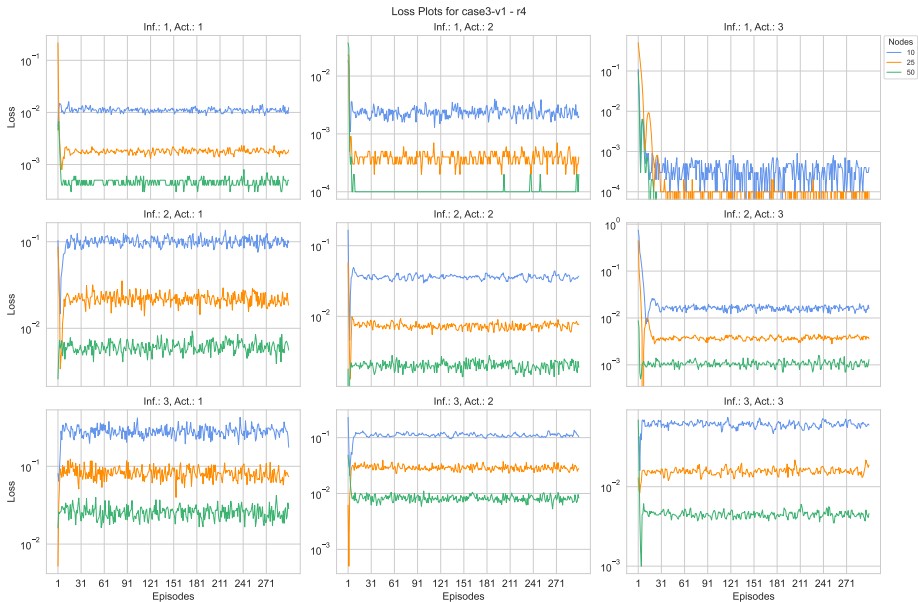

Figure 45: Case-3 v1 using R4: Training MSE loss evolution for RL policy using ResNet model across networks of 10 (blue), 25 (orange), and 50 (green) nodes, for varying initial misinformation sources (Inf.) and action budgets (Act.). Plotted on a logarithmic scale, the loss decreases over episodes, indicating improved policy performance and adaptation across network sizes.

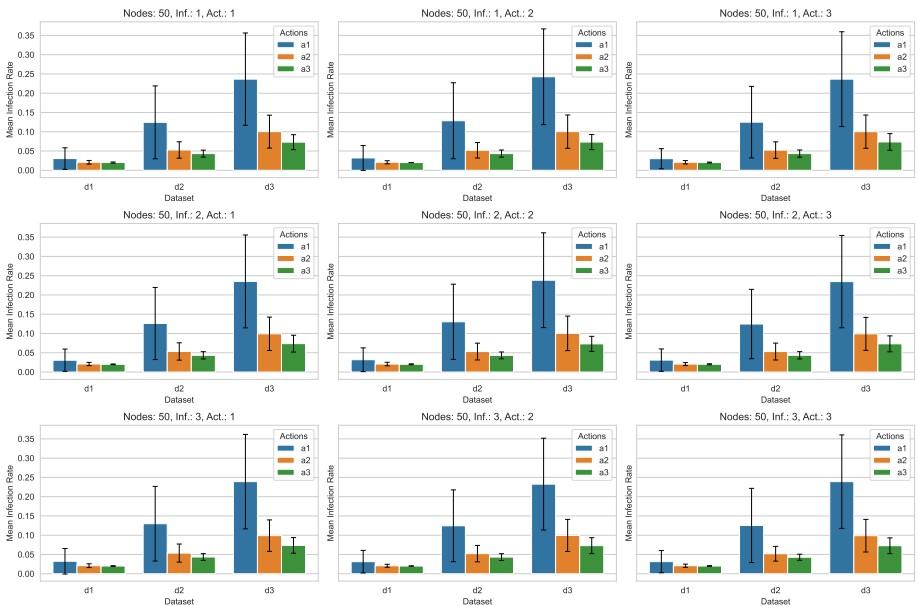

Figure 46: Case-3 v1 using R4: Performance comparison of the RL policies trained using ResNet model for a 50-node network. The barplot displays the mean infection rate for datasets d1, d2, and d3 of Dataset v1 type, differentiated by the number of initial misinformation sources (Inf.) and action budgets (Act.: a1, a2, a3). Each subplot illustrates the performance of a policy trained with the parameters indicated in its title. Lower infection rates indicate more effective policy learning and misinformation containment.

**Dataset v2 Results**

**Degree of connectivity 1**   50 Nodes - Figure 47

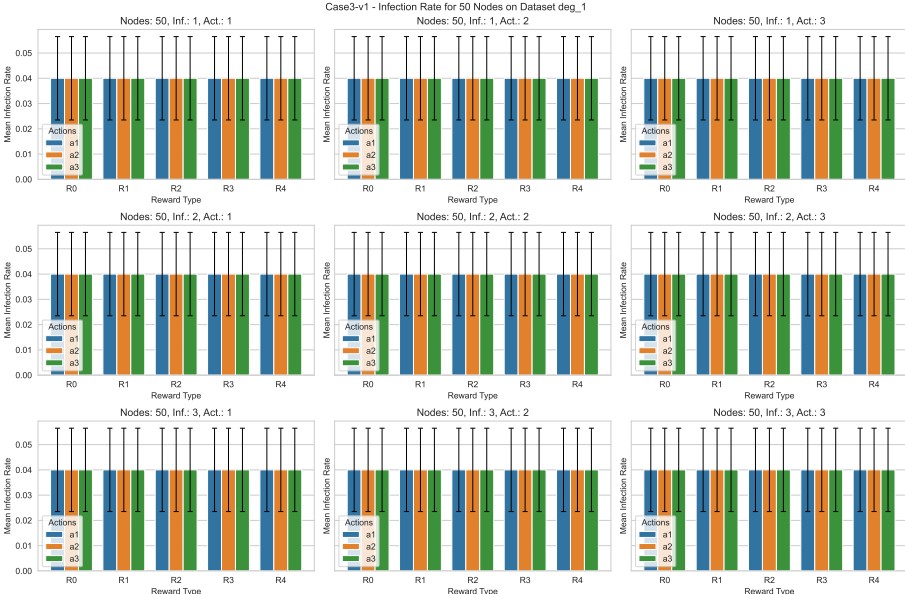

Figure 47: Case-3-v1 inference on Dataset v2: Performance comparison of the RL policies trained using ResNet model for a 50-node network. The barplot displays the mean infection rate for different reward functions on Dataset v2 of Degree of Connectivity 1, differentiated by the number of initial misinformation sources (Inf.) and action budgets (Act.: a1, a2, a3). Each subplot illustrates the performance of a policy trained with the parameters indicated in its title. Lower infection rates indicate more effective policy learning and misinformation containment.

**Degree of connectivity 2**    50 Nodes - Figure 48

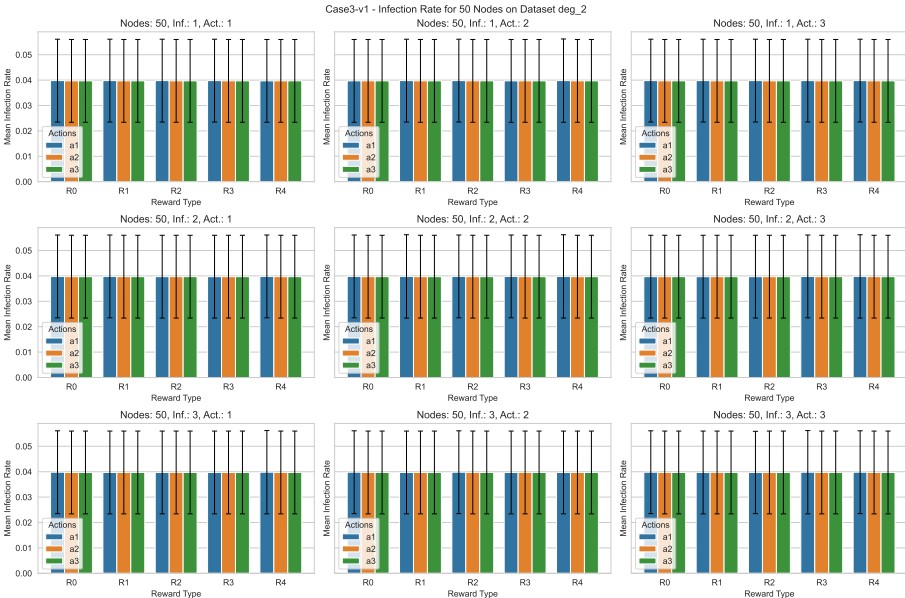

Figure 48: Case-3-v1 inference on Dataset v2: Performance comparison of the RL policies trained using ResNet model for a 50-node network. The barplot displays the mean infection rate for different reward functions on Dataset v2 of Degree of Connectivity 2, differentiated by the number of initial misinformation sources (Inf.) and action budgets (Act.: a1, a2, a3). Each subplot illustrates the performance of a policy trained with the parameters indicated in its title. Lower infection rates indicate more effective policy learning and misinformation containment.

**Degree of connectivity 3**   50 Nodes - Figure 49

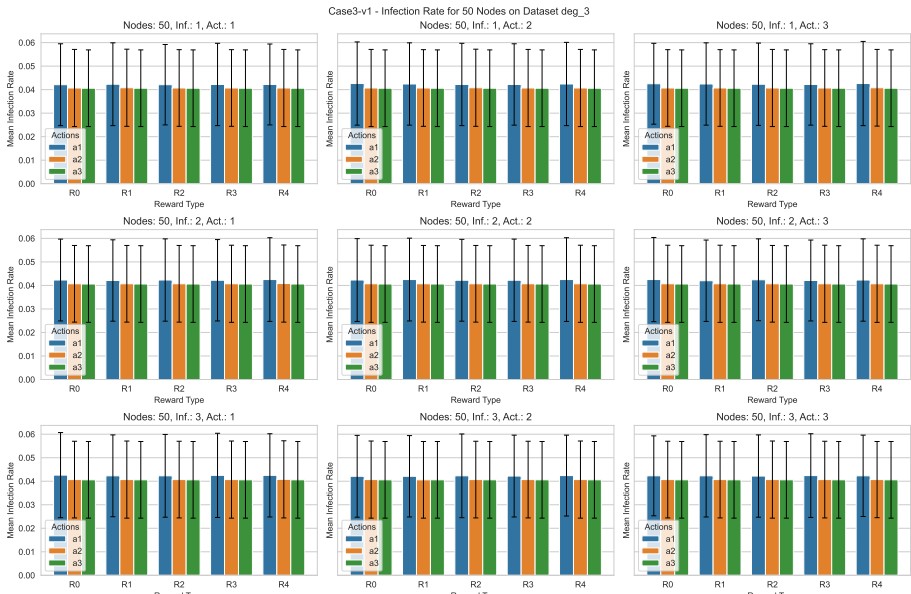

Figure 49: Case-3-v1 inference on Dataset v2: Performance comparison of the RL policies trained using ResNet model for a 50-node network. The barplot displays the mean infection rate for different reward functions on Dataset v2 of Degree of Connectivity 3, differentiated by the number of initial misinformation sources (Inf.) and action budgets (Act.: a1, a2, a3). Each subplot illustrates the performance of a policy trained with the parameters indicated in its title. Lower infection rates indicate more effective policy learning and misinformation containment.

**Degree of connectivity 4**    50 Nodes - Figure 50

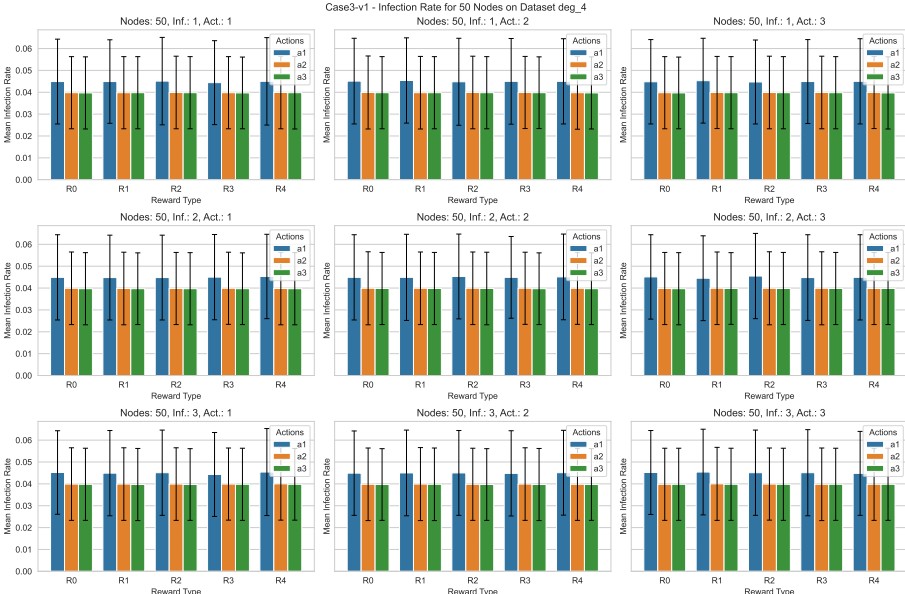

Figure 50: Case-3-v1 inference on Dataset v2: Performance comparison of the RL policies trained using ResNet model for a 50-node network. The barplot displays the mean infection rate for different reward functions on Dataset v2 of Degree of Connectivity 4, differentiated by the number of initial misinformation sources (Inf.) and action budgets (Act.: a1, a2, a3). Each subplot illustrates the performance of a policy trained with the parameters indicated in its title. Lower infection rates indicate more effective policy learning and misinformation containment.

