# OpenReview forum: "Towards Effective Planning Strategies for Dynamic Opinion Networks"
_NeurIPS.cc/2024/Conference — NeurIPS 2024 poster_

### Official Review · Reviewer_BrMA · 2024-06-20

**Soundness:** 4
**Presentation:** 4
**Contribution:** 3
**Rating:** 8
**Confidence:** 3

**Summary:**

This manuscript investigates intervention planning, a critical issue in complex social networks. To address this challenge, the authors introduce a novel ranking algorithm and a reinforcement learning-based dynamic planning framework. Three cases of opinion and trust values are considered, enhancing the method's applicability in real-world scenarios. The performance of the developed method is demonstrated through comprehensive simulations on synthetic networks.

**Strengths:**

This manuscript is well-written and engaging. The contributions of this work and the rationale for considering different models and factors are clearly illustrated. Various scenarios are accounted for to demonstrate the method's superiority and applicability. Four research questions are reasonably addressed in the manuscript. Overall, this work is novel and interesting and deserves to be published in NeurIPS.

**Weaknesses:**

One concern I have is the limited size of networks analysed in the simulation. As stated by the authors, “as network size increases, the problem becomes computationally intractable” (Line 6). Why do the authors only consider synthetic networks with fewer than 50 nodes in the simulation? Does this limitation affect the potential application of this method in real-world scenarios? Please provide reasonable explanations for this point. Additionally, the performance should be compared with state-of-the-art (SOTA) methods.

**Questions:**

Additional questions and suggestions:
1.	The authors mention that intervention planning involves two parts. However, it is not entirely clear how “exerting control” is considered in this work. Does it only mean adding green nodes in Figure 1? Please explain this more clearly.
2.	Only synthetic networks are considered to evaluate the developed method's performance, which could limit its applicability. The authors should consider some real-world networks, even without real-time opinion propagation data, to better demonstrate the method's superiority.
3.	In future work, real-time opinion propagation data should be considered, which could further enhance this study. This is not a critique of the current work but a suggestion for future research.
4.	A significant limitation is the size of networks analysed. The largest network contains only 50 nodes, which could impact the evaluation. Real-world networks typically have more than one million users (or at least more than one thousand users). The authors should explain how this method can be applied to real-world scenarios and justify why only networks with fewer than 50 nodes were analysed.
5.	What is the time complexity of this method? Is it difficult to analyse large-scale networks? Please provide an analysis of its time complexity and the resources needed for simulation.
6.	Can this method be compared with SOTA methods?
7.	Is Figure 19 correct? All bars and subfigures appear to be the same. Please verify this.

**Limitations:**

The authors have adequately addressed the limitations of their work.

---

> ### Author Rebuttal · Authors · 2024-08-07
>
> Thank you for recognizing our contributions and acknowledging our effort to cover various scenarios in our evaluation. We are also thankful for the constructive feedback and suggestions for future work.
>
> Yes, Figure 19 is correct. The barplot displayed in Figure 19 depicts the mean infection rate for different reward functions on Dataset v2 of Degree of connectivity 1, differentiated by the number of initial misinformation sources (Inf.) and action budgets (Act.: a1, a2, a3). We observe an identical average infection rate in all cases because each model successfully stops the spread of misinformation in the first timestep. This can be attributed to low degree of connectivity (one) of the initial infected node.
>
> **Remark 1, Question 2 & Question 4: Scalability and results from real-world networks**
> In our work we develop planning algorithms and analyze their efficacy using synthetic data in controlled settings. The synthetic data used in our study are generated using Watts-Strogatz (or small-world) network model. As demonstrated in various studies, this network model captures the characteristics of common social and community networks [1]. Additionally, we have also evaluated our planning algorithms using directed and undirected real-world network models reported in the literature. These evaluations are included in the rebuttal PDF (Table 2). These additional experiments confirm the major findings reported in our paper (Research Questions in Section 5).
>
> Furthermore, the Graph Convolutional Networks (GCNs)-based planners developed in our work perform effectively when applied to larger dynamic network models (as illustrated in Table 1 of our paper and Table 2 of rebuttal PDF) even when they are trained using data generated from smaller networks (RQ4 from Section 5). Testing with GCNs on larger networks is significantly simpler and less computationally intensive than training, making it a practical solution for scaling our approach to handle extensive network sizes efficiently. Finally, we would like to emphasize that existing research work [eg., citation 14 from our manuscript] has analyzed opinion networks with a maximum of 1005 nodes with an opinion propagation model that considers only binary opinion and trust values (Case 1 in our paper in Section 2.1). In contrast, we have also considered more expressive opinion network models (Case 2 and Case 3 -- Section 2.1) in our study.
>
> **Question 1: Details of World Model**
> In our work, "exerting control" refers to applying interventions (disseminating accurate information) for a selected node within the network to combat misinformation. Specifically, this involves adding an external input to the propagation model. For example, when agent $k$ communicates with agent $i$, agent $i$’s opinion is updated using the propagation model given in Equation 1 from our paper. If the planner chooses agent $i$ as the candidate node for intervention, exerting control implies an external input applied to this model and it will be governed by the following discrete map -
>
> $x_i(t+1) = x_i(t) + \mu_{ik}(x_k(t) - x_i(t)) + \mu_{ij}(x_j(t) - x_i(t))$
>
> $\mu_{ij}$ is the trust value associated with the external source $j$ sharing official information. $x_j$ is the associated opinion value.
>
> **Question 5: Time Complexity Analysis**
> The time complexities for the two main components of our method—the Ranking Algorithm and the Deep Value Network (DVN)—are as follows:
>
> Ranking Algorithm (Algorithm 1 in Appendix Section A.5.1): The time complexity is $O(V \times (V + E))$, where $V$ represents the number of vertices and $E$ is the number of edges in the network.
>
> Deep Value Network (DVN) (Algorithm 2 in Appendix Section A.5.2): The overall complexity of the DVN with experience replay is $O(e_{\text{max}} \times (V \times (V + E) + nm))$, where $e_{\text{max}}$ is the maximum number of episodes, $n$ is the batch size, and $m$ represents the computational complexity of operations within the neural networks.
>
> Analyzing large-scale networks with our Ranking Algorithm and Deep Value Network (DVN) can be computationally demanding. The quadratic dependency on the number of nodes $V$ and the involvement of all edges $E$ in the Ranking Algorithm, and the complex operations of the DVN, make the method resource-intensive. As specified earlier, we find that GCNs trained with data generated from smaller networks are useful for intervention planning even when the size of the opinion network increases. Due to the richness of the dynamic models in Cases 2 and 3, intervention planning with these expressive opinion propagation models were computationally more demanding than in Case 1 (which are commonly studied in the literature).
>
> **Question 6: Comparison with existing methodologies**
> Most prior research has only addressed static network scenarios (primarily Case 1), without considering the rich dynamics of Cases 2 and 3 (see Section 2.1 for the distinction). This is the first study to incorporate such a rich dynamic nature of opinion networks in the study of intervention planning strategies. We have reported a comparative analysis of our Ranking Algorithm based Supervised Learning with three other planning strategies (random, max_degree_static, max_degree_dynamic, see Figures in Appendix section A.6.1) not just for Case 1 but also for Cases 2 and 3. In addition, we have also reported a comprehensive analysis (by varying degrees of initial infected nodes and action budgets) of our Reinforcement Learning-based Centralized Dynamic Planner across various reward models for all three cases. This includes results of static network scenarios and candidate nodes as reward models which are considered in the existing work [citation 14 from our paper].

---

> > ### Comment · Reviewer_BrMA · 2024-08-13
> >
> > Thanks for the authors' response. The authors have addressed most of my concerns.

---

### Official Review · Reviewer_VE7T · 2024-07-02

**Soundness:** 2
**Presentation:** 3
**Contribution:** 2
**Rating:** 5
**Confidence:** 4

**Summary:**

The paper investigates intervention planning aimed at disseminating accurate information within dynamic opinion networks using learning strategies. It introduces a novel ranking algorithm to identify key nodes for disseminating accurate information and develops a Reinforcement Learning (RL)-based dynamic planning framework. The framework is tested on networks governed by two propagation models, incorporating both binary and continuous opinion and trust representations. The experimental results indicate that the proposed strategies can enhance infection rate control, especially with increased action budgets.

**Strengths:**

1.Timely Topic: The paper addresses the significant issue of misinformation spread in social networks, which is a pressing problem in today’s digital age.
2.Thorough Analysis: The paper presents a solid and detailed analysis of the proposed methods, providing in-depth insights into their effectiveness and behavior under various conditions.

**Weaknesses:**

1.Lack of Innovation: The primary issue with the paper is the lack of significant innovation. The methodologies proposed are incremental improvements over existing approaches rather than groundbreaking new techniques.
2.Comparison with Existing Methods: The paper does not sufficiently compare its methods with existing state-of-the-art techniques. This makes it challenging to assess the true novelty and effectiveness of the proposed approaches.

**Questions:**

1.How does the proposed RL-based dynamic planning framework compare with other existing state-of-the-art methods for misinformation control in terms of both performance and computational efficiency?
2.Can the proposed methodologies be scaled effectively to much larger networks, and what are the computational requirements for such scaling?

**Limitations:**

Highlight Novelty: Clearly articulate the novel contributions of the paper. It would be beneficial to focus on what sets this work apart from existing research in the same area.

---

> ### Author Rebuttal · Authors · 2024-08-07
>
> Thank you for pointing out the novelty of the ranking algorithm and the thoroughness of the analysis presented in the paper. In the following, we address the specific concerns on novelty, comparisons with existing literature, and the question on scalability.
>
> **Remarks 1 & 3: Novelty**
> Existing research on planning problems in the context of opinion networks often overlooks key features of opinion propagation models, such as their rich network dynamics, asynchronous communication, and the impact of factors like the degree of infected nodes, action budget, and various reward models on the effectiveness of planners. Our work addresses this gap by analyzing three distinct cases of opinion network models (Section 2.1)—ranging from binary to continuous spectra of opinion and trust values—and asynchronous communication (Section 2.2). This approach introduces greater richness and expressiveness to opinion dynamics models, making the model-based analysis of opinion propagation more reflective of real-world scenarios. Furthermore, we develop comprehensive datasets with a wide range of Watts-Strogatz (or small-world) network topologies (Section 4.2), varying degrees of initial infected nodes, action budgets, and reward models—from those requiring local network information to those utilizing global real-time network states (Section 3.2.1).
>
> From an algorithmic perspective, we extend the static ranking algorithm based on the classical BFS to the problem of centralized planning in dynamic opinion networks. Supporting supervised learning-based planners, this algorithm can help planners when the network size is small and the global network information is available. To understand planning strategies for large-networks, we also investigate reinforcement-learning-based centralized planners. In particular, our study explores five distinct reward structures, enhancing our understanding of reward dynamics and their computability (based on access to local vs global network information) that may be applicable to diverse applications depending on network observability and size. Further, the use of Deep Value Networks (DVN) proves more appropriate for our chosen application compared to the traditional and popular architecture of Deep Q-Networks (DQN), which has been investigated to study planning problems for opinion propagation models with simple binary state and trust parameters. DVN-based planners studied in our work are better suited for environments where the number of intervention nodes (action budget) is not fixed and varies dynamically. This flexibility allows DVN to handle dynamic and large-scale networks with greater efficiency as shown in our work  [see citation 14 from the manuscript].
>
> This makes our work a significant and non-trivial contribution to the existing literature on this topic.
>
> **Remark 2 & Question 1: Comparison with existing methodologies**
> Most prior research has only addressed static network scenarios (primarily Case 1), without considering the rich dynamics of Cases 2 and 3 (see Section 2.1 for the distinction). This is the first study to incorporate such a rich dynamic nature of opinion networks in the study of intervention planning strategies. We have reported a comparative analysis of our Ranking Algorithm based Supervised Learning with three other planning strategies (random, max_degree_static, max_degree_dynamic, see Figures in Appendix section A.6.1) not just for Case 1 but also for Cases 2 and 3. In addition, we have also reported a comprehensive analysis (by varying degrees of initial infected nodes and action budgets) of our Reinforcement Learning-based Centralized Dynamic Planner across various reward models for all three cases. This includes results of static network scenarios and candidate nodes as reward models which are considered in the existing work [citation 14 from our paper].
>
> **Question 2: Scalability**
> We have analyzed the performance of the developed planners as the number of opinion network nodes increase. In this context, the GCN-based planners developed in our work perform effectively when applied to larger dynamic network models (as illustrated in Table 1 of our paper and Table 2 of rebuttal PDF) even when they are trained using data generated from smaller networks (RQ4 from Section 5). We would like to emphasize that existing research work [eg., citation 14 from our manuscript] has analyzed opinion networks with a maximum of 1005 nodes with an opinion propagation model that considers only binary opinion and trust values (Case 1 in our paper in Section 2.1). In contrast, we have also considered more expressive opinion network models (Case 2 and Case 3 -- Section 2.1) in our study.
>
> **_Computational Requirements_**
> To answer the computational requirement for scalability we consider the computational intensity of our algorithms. The Ranking Algorithm (Algorithm 1 in Appendix Section A.5.1) has a time complexity of $O(V \times (V + E))$, which increases quadratically with the number of vertices and edges, making it computationally intensive for larger networks. The training of DVN (Algorithm 2 in Appendix Section A.5.2) has a time-complexity of $O(e_{\text{max}} \times (V \times (V + E) + nm))$, where $e_{\text{max}}$​ is the maximum number of episodes, $n$ is the batch size, and $m$ represents the computational complexity of operations within the neural networks. This indicates substantial computational requirements due to the need for simulating network dynamics across multiple training episodes and updating complex neural network parameters repeatedly.
>
> As specified earlier, we find that GCNs trained with data generated from smaller networks are useful for intervention planning even when size of the opinion network increases. Due to the richness of the dynamic models in Cases 2 and 3, intervention planning with these expressive opinion propagation models were computationally more demanding than Case 1 (which are commonly studied in the literature).

---

> ### Comment · Reviewer_VE7T · 2024-08-07
> **Accept**
>
> The rebuttal addressed most of my concerns, especially those regarding the scalability of the approach.

---

> > ### Author Response · Authors · 2024-08-08
> > **Thank you**
> >
> > Thank you for your comments. We are thankful that we could address your concerns.

---

### Official Review · Reviewer_jjEE · 2024-07-09

**Soundness:** 3
**Presentation:** 3
**Contribution:** 3
**Rating:** 7
**Confidence:** 4

**Summary:**

This paper studies strategic planning for disseminating credible information within dynamic opinion networks. The main two contributions are (1) a ranking algorithm to identify influential nodes to spread accurate information, and (2) an RL-based framework for adaptive intervention strategies.

**Strengths:**

1. Paper is well-written and easy to follow.
2. The problem at hand is interesting.

**Weaknesses:**

1. One of the main building blocks of the paper is a proposed ranking algorithm to mitigate the computationally intractable issue of intervention planning, while the algorithm itself is just a brute force search, and computationally infeasible. The authors then propose an RL solution to address the problem. My question is what did the authors achieve here?

2. My other concern is about the scalability of the solution as mentioned several times throughout the paper, making it one the main contributions of the paper, without providing further analysis that proves the proposed framework is indeed scalable. The largest network size used in the paper is 50, which makes it hard to believe the method is scalable, especially since their ranking algorithm is brute force (can we quantify scalability other than network size by the way?). I would recommend increasing network size to more than 50, while providing detailed statistics of the graphs (e.g., # edges, diameter of the graph, etc.). Furthermore, I assume the 1000 states used correspond to the dynamic nature of their input graphs. This has to be clarified in the paper.

3. Since the proposed method is claimed to be scalable, it is necessary to include analysis on how varying the number of states (or any other dynamic features of the input graphs) might affect the inference results.

4. Normally, one would expect the R2 reward function to achieve better results as it minimizes the # candidate nodes and infection rate, but as stated by RQ3, this isn’t the case. This requires a better explanation and justification other than just showing numbers in Table 1, especially because according to Table 1, R1,2 and 4 are quite close as far as I can see. Also, in general, except for R0 and R3, differences between others are insignificant. Some kind of significance test is thus recommended to clarify this better.

5. One last question is: does adding directions to the input graphs change anything in the paper? This needs to be shown in the paper using some further analysis and experiments.

6.  While the paper is very well-written, it still needs a thorough proof-read to fix some typos and mistakes (e.g., in the RQ3 section, Hypothesis RQ4 needs to be RQ3).

7. I’m not sure if I understood how the Infection rate=0.7 for figure 2 was calculated and compared to that of figure 1. I would recommend adding the infection rate for figure 1 somewhere and explain how to calculate this using an example.

**Questions:**

Please see the detailed comments that include some questions.

**Limitations:**

To some extent. Again, please see the questions above as they pose some challenges and limitations.

---

> ### Author Rebuttal · Authors · 2024-08-07
>
> Thank you for your valuable feedback. Thank you for pointing out that the paper is well written and for your valuable suggestions for improving our paper.
>
> With regards to the infection rate calculation - it is calculated as the ratio of the number of infected nodes to the total number of nodes within the network at a given timestep, as mentioned in Section A.3. We will clarify this in the main paper, and add this to Figure 1 for consistency.
>
> We will also thoroughly proof-read our paper to avoid typographical errors.
>
> **Remark 1: Building Blocks**
> Our paper focuses on two methodologies: Supervised Learning (SL) and Reinforcement Learning (RL). We aimed to extend SL to dynamic network scenarios (Case 2 and 3), which previous research works have not explored. To achieve this, we introduced a ranking algorithm based label generation approach within SL. As discussed in the paper (Section 3.1), while the ranking algorithm provided improvements, it proved computationally infeasible when training on larger networks. To address this, we investigated RL-based solutions (Section 3.2). Based on our analysis we observed that RL-based solution using GCN model performed better in generating effective plans for networks with more nodes even when they were trained with data generated from networks with fewer nodes (RQ4).
>
> In summary, we start by extending the static ranking algorithm based on the classical Breadth First Search to the problem of centralized planning in dynamic opinion networks. Supporting supervised learning-based planners, this algorithm can help planners when the network size is small and the global network information is available. To understand planning strategies for large networks, we also investigated reinforcement-learning-based centralized planners. We have reported comprehensive analyses in our work that were helpful in addressing specific research questions (Section 5 RQ1 to RQ4), which were not previously reported in the literature.
>
> **Remark 2, 3 & 5: On Scalability and Directed graphs**
> Following the reviewers' suggestions, we have included additional analysis in the attached PDF. These include results on four real-world network models used in previous works to analyze intervention planning. These include both directed and undirected networks with the number of nodes ranging from 34 to 2000. These experimental results align with the major findings reported in our paper (Research Questions in Section 5).
>
> The synthetic data used in our study is based on Watts-Strogatz (or small-world) network topology, which, as demonstrated in various studies, captures the characteristics of many common social and community networks [1]. The network statistics for this model are governed by the Watts-Strogatz network parameters, which are given in Section 4.2 of our paper. The network statistics of the four real-world networks considered are reported in Table 2 of the attached PDF. Further, we have considered two dataset versions (Section 4.2) that account for varying number of infected nodes and varying degree of initial infected nodes. In total we have 21 unique datasets and 1000 samples were randomly generated in each of these datasets for testing.
>
> We have analyzed the performance of the developed planners as the number of opinion network nodes increase. In this context, the GCN-based planners developed in our work perform effectively when applied to dynamic network models with more nodes (as illustrated in Table 1 of our paper and Table 2 of rebuttal PDF) even when they are trained using data generated from networks with fewer nodes (RQ4 from Section 5). We would like to emphasize that existing research work [eg., citation 14 from our manuscript] has analyzed opinion networks with a maximum of 1005 nodes with an opinion propagation model that considers only binary opinion and trust values (Case 1 in our paper in Section 2.1). In contrast, we have also considered more expressive opinion network models (Case 2 and Case 3 -- Section 2.1) in our study. Due to the richness of the dynamic models in Cases 2 and 3, intervention planning with these expressive opinion propagation models was computationally more demanding than Case 1.
>
> Thus, with regard to scalability, in our work, we mainly focus on the performance of centralized planners when the number of nodes increases. Varying the number of node features or the dimension of the opinion propagation model will introduce factors such as multiple topics and topic-dependencies in opinion networks. Though this is an important problem, this is beyond the current scope and will be investigated in our future study.
>
> **Remark 4: On Reward Models**
> Thank you for highlighting this intriguing result, which challenges the intuitive expectation that the R2 reward function would outperform others by minimizing candidate nodes and infection rate.
>
> *Global vs. Local Information:* As highlighted in RQ3 of our paper, our findings show that reward functions utilizing global information do not necessarily benefit from the addition of local information. This is evident with R2, which incorporates both global and local data but does not significantly outperform others using solely global metrics. The same is evident from Figure 7 in our appendix, which compares the MSE loss during training across different reward functions. **R2 (Green)** shows more variance and higher MSE loss, suggesting less stability and efficiency in learning compared to R1, R3, and R4.
>
> *Reward Richness and Application:* Although the differences in results for R1, R2, and R4 may appear insignificant, each reward type's computation and informational needs are distinct. While R4 requires the observability of the entire network, R1 only requires the observability of neighboring candidate nodes of the infected nodes. This diversity allows us to cater to various real-world applications, underscoring the necessity of evaluating multiple reward models to gain a comprehensive understanding.

---

> > ### Comment · Reviewer_jjEE · 2024-08-09
> > **Accept**
> >
> > Thank you for the detailed rebuttal that addressed most of my concerns.

---

> > > ### Author Response · Authors · 2024-08-09
> > >
> > > Thank you for your comments. We are thankful that we could address your concerns.

---

### Official Review · Reviewer_Z3oY · 2024-07-12

**Soundness:** 3
**Presentation:** 3
**Contribution:** 3
**Rating:** 6
**Confidence:** 3

**Summary:**

This study explores intervention strategies aimed at curbing the spread of misinformation in dynamic social networks. The authors propose a novel ranking algorithm to identify influential nodes for disseminating accurate information and a RL-based framework to address the computational complexity associated with label generation. The paper concludes that by integrating more realistic and complex modeling approaches, label generation techniques, and training methodologies, the proposed strategies can significantly mitigate the impact of misinformation in social networks.

**Strengths:**

- The studied problem is of great importance and has considerable practical significance.
- The paper introduces a novel ranking algorithm that effectively identifies key nodes within a network for the dissemination of accurate information. This algorithm is integrated with a supervised learning framework, providing a robust method for training neural network classifiers that can scale and generalize across different network structures.
- By employing a reinforcement learning-based dynamic planning framework, the paper addresses the computational challenges associated with large networks. This RL methodology allows for the development of adaptive intervention strategies that can respond in real-time to the evolving patterns of misinformation spread, offering a significant improvement over traditional static approaches.
- The authors have made the code available and provided detailed hyperparameter settings and computational resource information, enhancing the reproducibility of the study.

**Weaknesses:**

- While the proposed models and algorithms demonstrate efficacy in controlled settings, there may be concerns regarding their scalability when applied to real-world scenario.

**Questions:**

Why use ResNet and GCN as backbones?

**Limitations:**

The authors adequately addressed the limitations and broader impacts in the conclusion section.

---

> ### Author Rebuttal · Authors · 2024-08-07
>
> Thank you for acknowledging the technical contributions and novelty of our work and for providing valuable feedback.
>
> **Remark 1: Real-world network models**
> In our work, we develop planning algorithms and analyze their efficacy using synthetic data in controlled settings. The synthetic data used in our study are generated using the Watts-Strogatz (or small-world) network model. As demonstrated in various studies, this network model captures the characteristics of common social and community networks [1]. Additionally, we have also evaluated our planning algorithms using directed and undirected real-world network models reported in the literature. These evaluations are included in the rebuttal PDF (Table $2$). These additional experiments confirm the major findings reported in our paper (Research Questions in Section 5).
>
> Furthermore, the Graph Convolutional Networks (GCNs)-based planners developed in our work perform effectively when applied to dynamic network models with more nodes (as illustrated in Table $1$ of our paper and Table $2$ of rebuttal PDF) even when they are trained using data generated from networks with fewer nodes (RQ4 from Section 5). Testing with GCNs on larger networks is significantly simpler and less computationally intensive than training, making it a practical solution for scaling our approach to handle extensive network sizes efficiently. Finally, we would like to emphasize that existing research work [eg., citation 14 from our manuscript] has analyzed opinion networks with a maximum of $1005$ nodes with an opinion propagation model that considers only a binary opinion and trust values (Case 1 in our paper in Section 2.1). In contrast, we have also considered more expressive opinion network models (Case 2 and Case 3 -- Section 2.1) in our study.
>
> **Question 1: Usage of ResNet and GCN**
> Graph Convolutional Networks (GCNs) and Residual networks (ResNet) have been widely used for modeling graph-structured data. Since opinion dynamic models generate graph-structured data, we incorporated these neural network models in our study to develop planners. In our analysis, we found that GCNs offer better scalability and performance when compared with ResNet (RQ4 from Section 5).
>
> [1]. Watts, D.J. and Strogatz, S.H., 1998. Collective dynamics of ‘small-world’networks. nature, 393(6684), pp.440-442.

---

> > ### Comment · Reviewer_Z3oY · 2024-08-12
> >
> > Thank the authors for their response. I will keep my score.

---

> > > ### Author Response · Authors · 2024-08-12
> > >
> > > Thank you for your response. We wanted to check if our rebuttal addressed the raised concerns. We will be happy to address specific questions.

---

### Author Rebuttal · Authors · 2024-08-07

We thank the reviewers for their valuable feedback and suggestions. In this rebuttal, we have tried to address all the specific concerns and comments of individual reviewers. To support our response to the reviewers' comments, we include two additional tables in the attached PDF that are not part of the manuscript. These tables are intended to reinforce key findings, provide supporting evidence for our claims, and help clarify some of the reviewers' comments addressed in our rebuttal.

Table 1, in the attached Rebuttal PDF, highlights the key features of our work and their implications, emphasizing aspects not covered in previous studies related to planning in the context of opinion networks available in the literature. Existing research on planning problems in the context of opinion networks often overlooks key features of opinion propagation models, such as their rich network dynamics, asynchronous communication, and the impact of factors like the degree of infected nodes, action budget, and various reward models on the effectiveness of planners. Our work addresses this gap by analyzing three distinct cases of opinion network models (Section 2.1)—ranging from binary to continuous spectra of opinion and trust values—and asynchronous communication (Section 2.2). This approach introduces greater richness and expressiveness to opinion dynamics models, making the model-based analysis of opinion propagation more reflective of real-world scenarios. Furthermore, we develop comprehensive datasets with a wide range of Watts-Strogatz (or small-world) network topologies (Section 4.2), varying degrees of initial infected nodes, action budgets, and reward models—from those requiring local network information to those utilizing global real-time network states (Section 3.2.1). The small-world network topology subsumes some of the common social and community networks as shown in various studies [1]. As a result, experiments on four of the real-world network models (Rebuttal PDF - Table 2) (directed and undirected), previously studied in the literature, align with the major findings reported in our manuscript (Section 5 - Research Questions). This makes our work a significant and non-trivial contribution to the existing literature on this topic.

From an algorithmic perspective, we extend the static ranking algorithm based on the classical Breadth First Search to the problem of centralized planning in dynamic opinion networks. Supporting supervised learning-based planners, this algorithm can help planners when the network size is small and the global network information is available. To understand planning strategies for large-networks, we also investigated reinforcement-learning-based centralized planners. In particular, our study explores five distinct reward structures, enhancing our understanding of reward dynamics and their computability (based on access to local vs global network information) that may be applicable to diverse applications depending on network observability and size. Further, the use of Deep Value Networks (DVN) proves more appropriate for our chosen application compared to the traditional and popular architecture of Deep Q-Networks (DQN), which has been investigated to study planning problems for opinion propagation models with simple binary state and trust parameters. DVN-based planners studied in our work are better suited for environments where the number of intervention nodes (action budget) is not fixed and varies dynamically. This flexibility allows DVN to handle dynamic and large-scale networks with greater efficiency as shown in our work [see citation 14 from the manuscript].

[1]. Watts, D.J. and Strogatz, S.H., 1998. Collective dynamics of ‘small-world’networks. nature, 393(6684), pp.440-442.

---

### Decision · Program_Chairs · 2024-09-25

**Decision:**

Accept (poster)

**Comment:**

The paper studies strategic planning for disseminating accurate information within dynamic opinion networks. The authors propose a novel ranking algorithm and a reinforcement learning-based dynamic planning framework. The reviewers recognize the importance of mitigating misinformation spread in social networks and appreciate the thorough analysis of the results and clear paper presentation. The reviewers are unanimously positive after the rebuttal. The authors should incorporate the discussion and new results from the rebuttal into the final version. I recommend acceptance of the paper.